# Benchmarking the Limits of
# In-Context Reinforcement Learning for Ad-Hoc Teamwork

Yuheng Jing [1 2]   Kai Li [† 1 2]   Jiajun Zhang [3 4]   Zeyao Ma [4]   Jiaxi Yang [5]   Lei Zhang [5]   Zhe Wu [6]   Jinmin He [1 2]
Junliang Xing [6]   Jian Cheng [1 7 8]

## Abstract

In-Context Reinforcement Learning (ICRL) has enabled foundation agents to adapt instantaneously to novel tasks, yet its efficacy in Ad-Hoc Teamwork (AHT)—where coordination with unknown partners is required—remains unexplored. To rigorously evaluate this, we introduce a large-scale benchmark **ICRL4AHT**, built upon a high-throughput JAX implementation of Overcooked-V2. Our benchmark includes a large, diverse teammate suite spanning both RL and heuristic policies, enabling controlled train-test shifts, and provides a reproducible end-to-end pipeline for teammate generation, learning-history collection, dataset construction, and online multi-episode evaluation. We evaluate representative history-conditioned ICRL algorithms, including Algorithm Distillation (AD) and Decision-Pretrained Transformer (DPT), across millions of transitions. Results reveal notable limitations: contrary to their success in single-agent domains, these baselines fail to exhibit robust test-time adaptation in multi-agent settings. Specifically, these methods frequently underperform random baselines across both unseen teammate and unseen layout tracks, with no clear in-context improvement over long horizons. These findings highlight the challenges of strategic inference under partial observability within the OvercookedV2 AHT protocol, establishing our benchmark as a critical testbed for next-generation coordination algorithms.

## 1. Introduction

The paradigm of **In-Context Reinforcement Learning (ICRL)** has *fundamentally shifted* how agents adapt to novel environments (Moeini et al., 2025). By treating an agent's interaction history as a prompt for a sequence model, ICRL enables **rapid, few-shot adaptation** without the need for gradient updates or explicit fine-tuning. Recent foundational works (Lin et al., 2024; Wang et al., 2025) have demonstrated that algorithms such as Algorithm Distillation (AD) (Laskin et al., 2023) and Decision-Pretrained Transformer (DPT) (Lee et al., 2023) can achieve promising performance across diverse single-agent tasks, effectively *'learning to learn'* from context. This success suggests a promising path toward **foundation agents** capable of solving open-ended problems in dynamic environments.

However, the complexity of real-world deployment inherently involves *interaction with other autonomous entities* rather than static environments. This challenge is formalized as **Ad-Hoc Teamwork (AHT)**, where an agent must collaborate with novel partners *without prior coordination or communication protocols* (Knott et al., 2021). Theoretically, ICRL is an ideal candidate for AHT: an agent could *infer a partner's hidden intent or policy* purely from the interaction history—treating the teammate as a dynamic part of the environment context. Yet, unlike the existing literature which often relies on Population-Based Training (Li et al., 2023; Liang et al., 2024) or idealized self-play, the capability of general-purpose ICRL agents to adapt to **truly unseen, heterogeneous teammates** remains largely unverified.

Progress in this direction is currently stifled by the **lack of adequate evaluation infrastructure**. While benchmarks like XLand-100B (Nikulin et al., 2025) provide the scale necessary for single-agent ICRL, the multi-agent domain lacks a counterpart that offers both *scale (millions of transitions)* and *strategic diversity*. Existing AHT benchmarks typically rely on small sets of homogeneous partners or limited task variants (Carroll et al., 2019; Bard et al., 2020), failing to impose the severe distribution shifts required to rigorously test in-context generalization. Absent controlled variation in partner behavior across fixed tasks, observed performance *conflates task competence with partner-specific*

---

[1]C²DL, Institute of Automation, Chinese Academy of Sciences [2]School of Artificial Intelligence, University of Chinese Academy of Sciences [3]University of Science and Technology of China [4]Qwen Team, Alibaba Group [5]Shenzhen Institutes of Advanced Technology, Chinese Academy of Sciences [6]Department of Computer Science and Technology, Tsinghua University [7]School of Future Technology, University of Chinese Academy of Sciences [8]AiRiA. Correspondence to: Kai Li <kai.li@ia.ac.cn>.

*Proceedings of the 43^{rd} International Conference on Machine Learning*, Seoul, South Korea. PMLR 306, 2026. Copyright 2026 by the author(s).

*overfitting*, precluding any claim of genuine coordination.

To bridge this gap, we introduce **ICRL4AHT**, a large-scale benchmark with a reproducible dataset-generation and evaluation pipeline specifically designed to *probe the limits of ICRL in cooperative settings*. Built upon a high-performance JAX implementation of OvercookedV2 (Gessler et al., 2025), our framework features a highly efficient generation pipeline that curates a **Teammate Suite** of *unprecedented diversity*, comprising both diverse RL policies and distinct heuristic policies. We structure the benchmark around two rigorous evaluation tracks: (1) **Teammate Generalization**, testing adaptation to unseen behaviors within familiar layouts, and (2) **Layout Generalization**, testing robustness to novel environmental dynamics. This design ensures that high performance requires **genuine in-context reasoning** rather than statistical overfitting.

We utilize ICRL4AHT to conduct the first large-scale empirical analysis of representative history-conditioned ICRL methods (AD and DPT) in the context of AHT. Our results reveal a notable limitation in current sequence-model ICRL baselines: across both generalization tracks, existing ICRL methods **often underperform simple random baselines**. Furthermore, we observe **flat learning curves** during evaluation, indicating a near-absence of the expected in-context improvement, even as the interaction horizon expands. Further ablations on context length and teammate-action conditioning suggest that the gap is *not merely an implementation detail* or an easily tuned hyperparameter, but reflects a **substantive limitation** of the studied ICRL baselines when faced with realistic partner diversity and environment shifts.

We release the full benchmark, including the *teammate suite, dataset generator, and evaluation pipeline*, to facilitate reproducibility and future research. Our repository is available at `https://github.com/AHT-Hub/ICRL4AHT`.

Our contributions are threefold:

- **A Scalable Benchmark and Reproducible Pipeline**: We provide a *reproducible, JAX-based pipeline* for teammate generation, learning-history collection, dataset construction, and standardized evaluation of cooperative multi-agent ICRL methods with controllable diversity.

- **A Diverse Teammate Suite**: We construct a heterogeneous library of teammates that *strictly separates training (RL-based) and testing (heuristic-based) distributions*, enforcing a rigorous test of generalization.

- **Benchmarking the Limits**: We demonstrate that representative history-conditioned sequence-model ICRL baselines, while successful in single-agent settings, struggle with the strategic inference required for partner adaptation under the OvercookedV2 AHT protocol studied here, highlighting a critical new frontier for the community.

## 2. Background

### 2.1. In-Context Reinforcement Learning

Transformers have become a standard backbone for decision-making by reframing RL as *sequence modeling* (Chen et al., 2021; Janner et al., 2021; Brandfonbrener et al., 2022; Yang et al., 2023) and enabling scalable conditioning and multitask pretraining (Furuta et al., 2022; Paster et al., 2022; Villaflor et al., 2022; Lee et al., 2022; Reed et al., 2022; Jing et al., 2024a; 2025b). When pretrained with appropriate contextual structure, these models can exhibit ICRL—**gradient-free test-time adaptation** driven purely by conditioning on interaction histories—closely related to recurrent or meta-RL formulations (Wang et al., 2017; Duan et al., 2016); see Beck et al. (2025) for a comprehensive survey of meta-RL. Recent advances include noise-distillation-based emergence of ICRL (Zisman et al., 2024), scalable in-context agents such as AMAGO (Grigsby et al., 2024) and Vintix (Polubarov et al., 2025), and analyses of when Q-learning can outperform model-based alternatives in this regime (Tarasov et al., 2025). In AHT, this capability is especially desirable because an agent must *infer and respond to unseen partner behaviors* from limited online interaction; ICRL4AHT instantiates this evaluation setting with standardized online protocols and tracks that disentangle generalization across teammates and layouts.

**Algorithm Distillation (AD).** AD trains a sequence model to *autoregressively predict actions* from past observations, actions, and rewards (Laskin et al., 2023). A crucial ingredient for eliciting **in-context (rather than in-weights) learning** is that the context contains *multiple episodes ordered by increasing return*, which differs from DT-style return-conditioning (Chen et al., 2021). XLand-100B adopts AD as a canonical ICRL method due to its simplicity and its ability to subsume several later variants (Mirchandani et al., 2023; Liu & Abbeel, 2023; Raparthy et al., 2024; Shi et al., 2023). In our benchmark, AD serves as a primary baseline for testing whether sequence models can *infer and exploit teammate-specific regularities* from short online histories.

**Decision-Pretrained Transformer (DPT).** DPT is motivated by a *Bayesian-inference approximation* (Müller et al., 2022) and trains a Transformer to predict the **optimal action** at a query state given a task-specific context that *need not be ordered*, trading away full learning histories for expert-action supervision (Lee et al., 2023). Recent theoretical work suggests that AD-style or DPT-style models can implement *near-optimal online RL algorithms 'within the forward pass'*, strengthening their appeal as test-time adaptation mechanisms (Lin et al., 2024; Wang et al., 2025; Dong et al., 2025). ICRL4AHT explicitly supports this supervision channel via optional expert-action relabeling in dataset construction, enabling controlled evaluation of DPT-style ICRL in cooperative coordination settings.

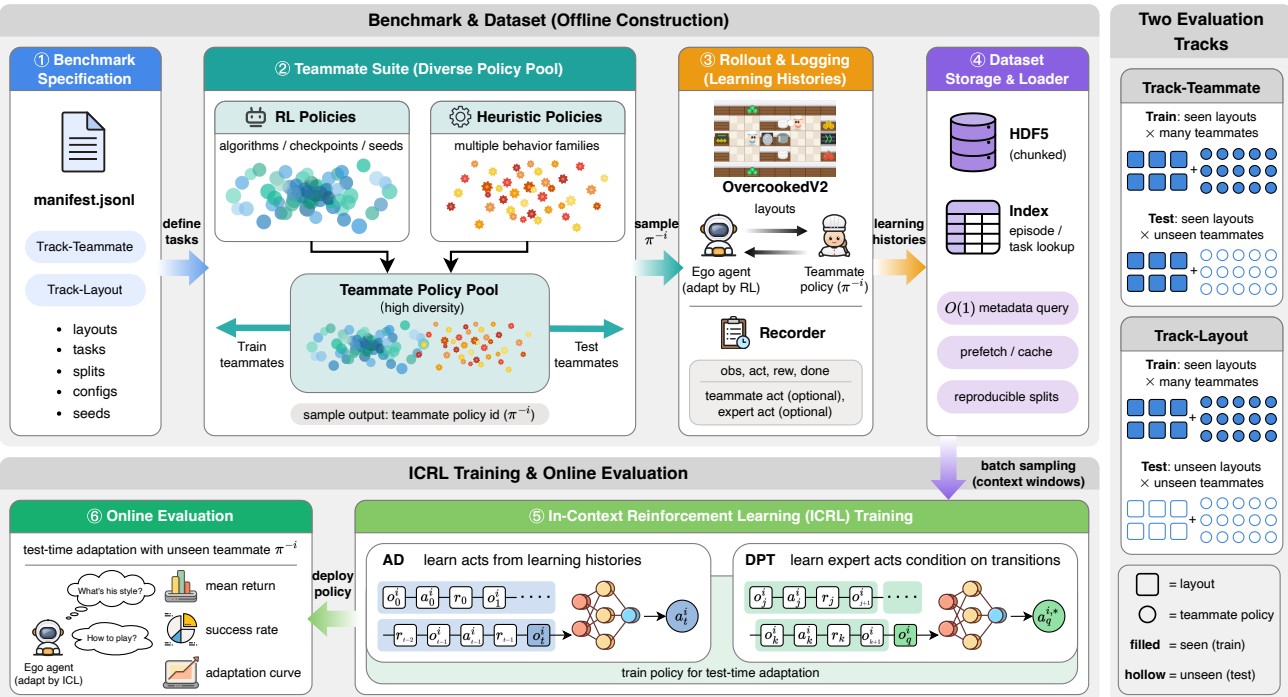

*Figure 1.* Benchmark Pipeline Overview (**ICRL4AHT**). (1) A benchmark manifest specifies layouts, tasks, and other properties, as well as defines two evaluation tracks that disentangle generalization over teammates and layouts. (2) A diverse teammate policy pool is constructed from both RL-trained policies and heuristic policies, where teammate policies $\pi^{-i}$ can be sampled for either training or testing. (3) Using OvercookedV2 layouts, interactions between an RL ego agent and a sampled teammate policy are rolled out and recorded as learning histories. (4) The collected data are packaged into a chunked HDF5 dataset with an episode / task index, fast metadata queries, and prefetch / cache support, enabling efficient context-window batch sampling for ICRL training. (5) ICRL learners are trained on the dataset and then (6) evaluated online for test-time adaptation under the two tracks, reporting metrics such as adaptation curves.

## 2.2. Ad-Hoc Teamwork

**Algorithmic Frameworks.** AHT studies cooperation in teams formed *without pre-coordination*, where an agent must coordinate with previously unseen partners under shared rewards (Stone et al., 2010). A dominant methodology is to train for **partner generalization** by exposing the learner to a diverse population of teammates, often built via Population-Based Training and then sampling partners from the population to induce multiple 'conventions' (Jaderberg et al., 2017; 2019). Concretely, prior work improves robustness by (1) *diversity-aware population learning* (Lupu et al., 2021; Zhao et al., 2023), (2) *partner modeling or inference* (Wu et al., 2021; Laidlaw & Dragan, 2022; Villin et al., 2025), (3) *domain randomization* (Tang et al., 2021; Yu et al., 2023), and (4) *quality-diversity style training* (Parker-Holder et al., 2020; Wu et al., 2023; Jing et al., 2025a).

These directions connect tightly to ZSC formalisms (Treutlein et al., 2021) and symmetry-breaking methods such as Other-Play (Hu et al., 2020) and Off-Belief Learning (Hu et al., 2021), as well as trajectory-diversity objectives for broadened convention coverage (Lupu et al., 2021). Despite progress, recent analyses argue that standard coordination benchmarks can *conflate coordination with out-*

*of-distribution state coverage*, and that even strong ZSC methods still struggle on scenarios that intrinsically require test-time protocol formation and adaptation (Gessler et al., 2025). This critique mirrors the observation that population-based approaches are often *task-specific* and offer *limited test-time adaptability* beyond what was seen in training partners. Motivated by this gap, ICRL4AHT reframes AHT as **in-context adaptation from interaction histories** (rather than re-training), and directly evaluates this axis using ICRL baselines on teammate or layout generalization.

**Benchmarks and Datasets.** Existing AHT benchmarks have largely been *environment-first*, with limited emphasis on standardized data assets and test-time adaptation protocols. Overcooked-AI (Carroll et al., 2019) popularized cooperative cooking for human-AI coordination and ZSC-style evaluation, and released a small human-human corpus ($\approx 16$ joint trajectories per layout). Recent OvercookedV2 variants further strengthen the coordination signal by introducing **meaningful partial observability** and **stochasticity**, mitigating self-play 'handshake' overfitting and making adaptation more necessary. Complementarily, Hanabi (HLE) (Bard et al., 2020) formalizes *implicit communication under information asymmetry*, and HOAD (Sarmasi et al., 2021) pro-

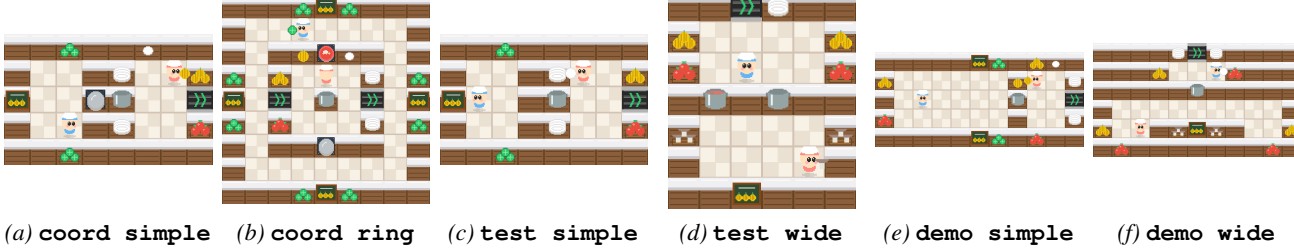

*(a)* `coord simple`    *(b)* `coord ring`    *(c)* `test simple`    *(d)* `test wide`    *(e)* `demo simple`    *(f)* `demo wide`

*Figure 2.* Representative OvercookedV2 layouts used in the ICRL4AHT benchmark. Each layout features distinct spatial configurations, imposing diverse coordination challenges. Two agents (blue-ego agent and red-partner) must collaborate under partial observability to prepare and deliver dynamically changing recipes. We supplement OvercookedV2's full environment documentation in Sec. A.

*Table 1.* **Comparison of AHT/MARL benchmarks vs. ICRL4-AHT.** Legend: ✓ explicitly supported; △ possible but not standardized as the primary artifact/protocol; – unsupported or N/A.

| Benchmark [B] / Dataset [D] (Domain) | Unseen Partners | Unseen Tasks/Layouts | Learning Histories | Data+Eval Pipeline |
|---|---|---|---|---|
| **ICRL4AHT [B,D]** (AHT, POMDP, ICRL) | ✓ | ✓ | ✓ | ✓ |
| Overcooked-AI [B,D] (AHT) | △ | △ | – | △ |
| OvercookedV2 [B] (AHT, POMDP) | ✓ | △ | – | △ |
| Hanabi (HLE) [B] (AHT, POMDP) | △ | – | – | – |
| Hanabi-HOAD [D] (Offline AHT) | △ | – | – | △ |
| Melting Pot 2.0 [B] (Social MARL Suite) | ✓ | ✓ | – | △ |
| OG-MARL [D] (Offline MARL) | – | △ | – | △ |

vides curated game logs (*e.g.*, 3,000+ high-quality matches) to train human-proxy policies for standardized evaluation against diverse human behaviors. Beyond small-team co-operation, Melting Pot 2.0 (Agapiou et al., 2022) evaluates 'stranger' generalization using background populations across 50+ substrates and 250+ scenarios, but is primarily an evaluation suite rather than a released dataset. Broader MARL resources such as GRF (Kurach et al., 2020) and LBF (Papoudakis et al., 2021) stress high-dimensional control and scalability. More broadly, offline MARL datasets such as OG-MARL (Formanek et al., 2023) standardize static experience ($\approx$ 1M transitions per task), which is valuable for data-centric MARL but not tailored to AHT adaptation. We summarize a compact comparison in Table 1.

Despite this rich ecosystem, ICRL-style AHT evaluation remains **under-served**: existing benchmarks typically (1) offer *limited controlled teammate diversity*, (2) *conflate teammate and environment generalization*, or (3) do not provide *learning-history datasets aligned with in-context adaptation*. ICRL4AHT is designed to close these gaps with an end-to-end benchmark pipeline, which provides (1) a **con-**trolled teammate suite with held-out partners**, (2) **disentangled teammate vs layout generalization tracks**, and (3) **learning-history datasets** and a standardized pipeline for ICRL-style in-context adaptation.

## 3. Problem Formulation

### 3.1. OvercookedV2 Environment as a Dec-POMDP

We model the cooperative multi-agent environment as a two-player Decentralized Partially Observable Markov Decision Process (Dec-POMDP) (Yang et al., 2025), defined by $\mathcal{M} = \langle \mathcal{S}, \mathcal{A}, P, R, \Omega, O, \gamma, T \rangle$. Here, $\mathcal{S}$ is the set of global states. $\mathcal{A} = \mathcal{A}^1 \times \mathcal{A}^2$ is the joint action space, where each agent $i \in \{1, 2\}$ chooses an action $a^i$ from a discrete set of interaction primitives. $P(s'|s, \mathbf{a})$ denotes the state transition dynamics given joint action $\mathbf{a} = (a^1, a^2)$. $R(s, \mathbf{a})$ is the shared reward function, representing the team's collective goal. $\Omega$ is the set of partial observations, and $O(o^{i'}|s', \mathbf{a})$ determines the observation received by agent $i$. Moreover, $\gamma \in [0, 1)$ is the discount factor, and $T$ is the horizon.

**OvercookedV2 Instance.** We instantiate $\mathcal{M}$ using OvercookedV2 (Gessler et al., 2025), which introduces *critical complexities* absent in the original Overcooked-AI. Unlike the fully observable predecessor, OvercookedV2 enforces **partial observability**, where $o^i$ represents a restricted field of view. The game is **highly stochastic**: agent starting positions are randomized at initialization, and the target recipe changes dynamically upon each successful delivery. Furthermore, reward function $R$ is *non-monotonic*; it yields positive sparse rewards for correct deliveries but *strictly penalizes incorrect ones*, thereby demanding **precise coordination** and continuous monitoring of the changing task context. We illustrate representative OvercookedV2 layouts in Fig. 2.

### 3.2. ICRL for Ad-Hoc Teamwork

We focus on the AHT setting from the perspective of a single controllable agent (the ego agent, $i$), adapting to a teammate (the partner, $-i$) with an unknown policy $\pi^{-i}$.

**Teammate as Context.** From the perspective of the ego

agent $i$, the partner $\pi^{-i}$ functions as an *intrinsic part of the environment*, inducing effective transition dynamics $P_{\pi^{-i}}(s'|s, a^i) = \sum_{a^{-i}} P(s'|s, a^i, a^{-i})\pi^{-i}(a^{-i}|o^{-i})$. While general AHT scenarios may involve non-stationary or co-adapting partners, we strictly enforce that $\pi^{-i}$ **remains fixed** throughout the entire interaction history. This constraint is critical to our definition of ICRL for AHT: it isolates the challenge of ***unilateral in-context adaptation***, requiring the ego agent to identify the partner's specific policy solely from historical cues ($h_t^i$) and adjust its behavior to maximize the joint return $R$ *without the confounding factor of the partner simultaneously adapting to the ego agent*.

**The ICRL Objective.** We adopt the ICRL paradigm, modeling the ego agent as a causal sequence model (*e.g.*, a Transformer) parameterized by $\theta$. At timestep $t$, the agent receives a context history $h_t^i$ containing the sequence of previous transitions within the context window $K$: $h_t^i = (\dots, o_{t-1}^i, a_{t-1}^i, r_{t-1}, o_t^i)$. The ego agent's policy is defined as $\pi_\theta(a_t^i|h_t^i) = \text{Transformer}_\theta(h_t^i)$. While both methods optimize a negative log-likelihood objective, they differ fundamentally in the data distribution $\mathcal{D}$ they consume:

- **AD** is trained on *learning histories* generated by some source RL algorithms interacting with teammates. A training sample consists of a sequence of episodes $\tau^i = (\tau_0^i, \dots, \tau_N^i)$ reflecting the source algorithms' learning progress *from random to expert*. AD minimizes:

$$\mathcal{L}_{\text{AD}}(\theta) = \mathbb{E}_{\tau^i \sim \mathcal{D}_{\text{learning}}}\left[-\sum_t \log \pi_\theta(a_t^i|h_t^i)\right]. \quad (1)$$

By modeling the improvement across episodes in $h_t^i$, AD learns to perform ***in-context meta-exploration***, identifying teammate and adapting its policy to maximize returns.

- **DPT** is trained on *expert trajectories* to perform **in-context task inference**. The objective is to predict the optimal action $a_t^{i,*}$ given a context $h_t^i$ that characterizes the current task dynamics (*i.e.*, the teammate's behavior):

$$\mathcal{L}_{\text{DPT}}(\theta) = \mathbb{E}_{\tau^{i,*} \sim \mathcal{D}_{\text{expert}}}\left[-\sum_t \log \pi_\theta(a_t^{i,*}|h_t^i)\right]. \quad (2)$$

Unlike AD, DPT does *not learn to explore*; instead, it treats $h_t^i$ as a prompt to identify the underlying MDP and directly **query the optimal policy**, effectively performing **zero-shot adaptation** to the identified teammate policy.

Let $\mathcal{L}$ represents the set of layouts and $\Pi$ represents the set of teammate policies, then the ultimate evaluation goal for both methods is to maximize the expected return over the distribution of tasks ($\mathcal{L} \times \Pi$) during online inference:

$$J(\theta) = \mathbb{E}_{l \sim \mathcal{L}, \pi^{-i} \sim \Pi}\left[\sum_{t=0}^{T-1} \gamma^t R(s_t, a_t^i, a_t^{-i})\Big| a_t^i \sim \pi_\theta(\cdot|h_t^i)\right].$$

## 4. The ICRL4AHT Benchmark

To rigorously probe the limits of in-context adaptation in multi-agent coordination, we introduce ICRL4AHT, a comprehensive benchmark designed for *scale*, *reproducibility*, and *disentangled evaluation*. The pipeline, shown in Fig. 1, bridges the gap between AHT and large-scale sequence modeling through three key stages: (1) specification and teammate generation, (2) high-throughput data collection and storage, (3) standardized ICRL training and evaluation.

### 4.1. Benchmark Specification & Teammate Suite

A central challenge in AHT evaluation is ensuring that test-time performance reflects *genuine adaptation* rather than overfitting to fixed partners. To address this, ICRL4AHT enforces *strict reproducibility* and *distribution shifts* through Manifest Specifications and a diverse Teammate Suite.

**Manifest Specification.** We decouple task definition from environment execution using *version-controlled JSONL manifests*. These manifests precisely define the evaluation tracks, layout configurations, and teammate policy assignments (*e.g.*, checkpoints, seeds) for both training and testing splits. This ensures that the **exact distribution of tasks is reproducible** across different compute infrastructures.

**Teammate Suite.** We construct a **heterogeneous library of teammate policies** comprising two complementary categories: RL-trained policies and heuristic policies. Both categories are designed to be interchangeable for either training or evaluation purposes, providing *flexible configurations* and a *wide spectrum of cooperative behaviors*. The full teammate policy generation details are provided in Sec. B.

- **RL Policies**: We train a diverse population of partners using a line of RL algorithms known for producing diversified teammate policies. This includes: (1) FCP (Strouse et al., 2021) which varys seeds and checkpoints, (2) BR-Div (Rahman et al., 2023) which optimizes *best response diversity metric*, (3) LBRDiv (Rahman et al., 2024) which emulates *minimum coverage set*, and (4) CoMeDi (Sarkar et al., 2024) which optimizes *mixed-play*.

- **Heuristic Policies**: We design four distinct heuristic teammate policy families with *varying levels of autonomous task-completion capability*, ordered by increasing cooperability: H1: `recipe_aware` < H2: `territory` < H3: `assembly_line` < H4: `utility_greedy`. The H1 family exhibits *minimal autonomous contribution*, requiring substantial coordination effort from the ego agent. The H2 family operates within designated spatial regions with limited flexibility. The H3 family follows structured sequential workflows, enabling moderate cooperation. The H4 family *autonomously pursues high-utility actions*, making coordination most tractable. This cooperability hierarchy enables **systematic evaluation** of ICRL

*Table 2.* **Dataset statistics per layout in ICRL4AHT.** We report key properties of the learning-history dataset we curated. Within, observation shape is $(5, 5, \star)$ where $\star$ varies by layout (See Fig. 2): 40 for (a,b,e), 41 for (c), 46 for (d), and 45 for (f).

| Property | Value |
|---|---|
| Original Transitions | 1,196,032,000 |
| Filtered Transitions | 149,504,000 |
| History Length | 14,600 |
| Num. Teammates | 80 |
| Steps per Episode | 100 |
| Obs. Shape | $(5, 5, \star)$ |
| Num. Actions | 6 |
| Mean Final Return | $\approx 40$ |
| Disk Size (compressed) | $\approx 6.5$ GB |

methods across a spectrum of coordination difficulty.

In our experiments, we deliberately employ **RL policies exclusively for training** and **heuristic policies exclusively for testing**, thereby inducing a *controlled distribution shift* that rigorously probes generalization abilities. As quantified in Sec. F.2, the heuristic families exhibit *high pairwise Hamming distances* from RL policies, confirming that adaptation to held-out families requires **genuine in-context inference** rather than interpolation within similar policy clusters.

### 4.2. Learning History Collection & Data Infrastructure

Standard offline RL datasets typically consist of static expert trajectories (Fu et al., 2020), which *fail to capture the process of adaptation*. To enable AD, we require **'learning histories'**—sequences of episodes where an agent improves *from random to expert behavior* against a specific teammate.

**Rollout and Logging.** We implement a **high-performance, JAX (Bradbury et al., 2018)-native parallel rollout engine** capable of generating *billions of transitions*. We train an ego PPO (Schulman et al., 2017) agent against each teammate in the training manifest, recording complete interaction histories $\{(o^i, a^i, a^{-i}, r)\}$. See Sec. C for training details. To enhance data quality, we apply *trajectory filtering* that selects high-quality learning curves based on final performance and improvement, ensuring the Transformer learns **genuine improvement dynamics**, which is detailed in Sec. F.1.

**Dataset Storage and Loader.** To handle the scale of the data efficiently, we serialize histories into a *chunked HDF5 format* optimized for contiguous memory access. The storage system is paired with a lightweight JSON index that allows $O(1)$ **metadata lookup** for arbitrary episodes. We provide the dataset structure details in Sec. D.

- **Scale**: As detailed in Table 2, resulting dataset comprises over **1.19B original transitions**, filtered down to $\approx 150M$ *high-quality transitions* across 80 unique teammates.

- **Efficiency**: The data loader supports *threaded prefetching*

*and smart caching*, maximizing GPU utilization during Transformer training by minimizing I/O bottlenecks.

### 4.3. ICRL Training & Online Evaluation Protocols

The final stage of the pipeline standardizes the training of sequence models and their evaluation in online interactions.

**ICRL Training.** The framework natively supports state-of-the-art ICRL architectures. We provide reference implementations for AD and DPT (as detailed in Sec. E). The dataset loader allows flexible context window sampling, enabling analysis of how context length affects the adaptation.

**Evaluation Tracks.** Building upon the training task distribution $\mathcal{D}_{\text{train}} = \mathcal{L}_{\text{train}} \times \Pi_{\text{train}}$, we build two evaluation tracks with **disjoint test sets** to rigorously assess generalization:

- **Track 1 (Teammate Generalization)**: This track probes adaptation to *held-out partners within familiar environments*. Test set is defined as $\mathcal{D}_{\text{test}}^{(1)} = \mathcal{L}_{\text{train}} \times \Pi_{\text{test}}$, where $\Pi_{\text{train}} \cap \Pi_{\text{test}} = \emptyset$. Success necessitates **rapid in-context identification** of partner intent and behavioral patterns.

- **Track 2 (Layout Generalization)**: This track assesses robustness under *severe distribution shift*, pairing novel partners with novel environments: $\mathcal{D}_{\text{test}}^{(2)} = \mathcal{L}_{\text{test}} \times \Pi_{\text{test}}$, where $\mathcal{L}_{\text{train}} \cap \mathcal{L}_{\text{test}} = \emptyset$. This setting imposes a **dual challenge**, requiring *simultaneous adaptation to novel environmental dynamics and unseen partner policies*.

## 5. Experiments

We evaluate the efficacy of state-of-the-art ICRL algorithms, specifically AD and DPT, on the ICRL4AHT benchmark. Our experiments are designed to answer two fundamental questions: (1) *Can ICRL agents generalize to unseen teammates with distinct behavioral policies?* (2) *Can they maintain coordination when both the teammate and the environment layout change simultaneously?*

### 5.1. Track 1: Teammate Generalization

In this track, agents trained on a diverse set of RL-based partners are evaluated against four held-out families of `H1–H4` on familiar layouts. The results, summarized in Table 3, reveal **significant limitations** in current ICRL approaches.

**Performance vs. Baselines.** Contrary to the success observed in single-agent domains, both AD and DPT **struggle to consistently outperform a random baseline** in cooperative settings. DPT achieves the highest average return $(12.4 \pm 11.0)$, marginally outperforming the Random baseline $(5.5 \pm 5.6)$ and AD $(9.1 \pm 5.8)$. However, this average is *heavily skewed by the `H4` family*, which is autonomously capable and requires minimal coordination. Statistical protocol: each teammate instance is evaluated over 100 episodes;

*Table 3.* **Track 1: Teammate Generalization** ($\mathcal{L}_{\text{train}} \times \Pi_{\text{test}}$). Performance comparison across six training layouts and four teammate families. Green indicates higher returns, red indicates negative returns. **Bold** marks the best method per layout. The studied ICRL baselines (AD, DPT) *fail to consistently outperform random baselines*, notably exhibiting **catastrophic failures** on `test wide`.

| Heuristic Family | coord simple | | | coord ring | | | test simple | | | test wide | | | demo simple | | | demo wide | | |
|---|---|---|---|---|---|---|---|---|---|---|---|---|---|---|---|---|---|---|
| | AD | DPT | Rnd | AD | DPT | Rnd | AD | DPT | Rnd | AD | DPT | Rnd | AD | DPT | Rnd | AD | DPT | Rnd |
| H1 | 0.0±0.0 | 0.0±0.0 | -4.6±0.2 | 0.0±0.0 | 0.0±0.0 | -1.3±0.2 | 0.0±0.0 | 0.0±0.0 | 0.0±0.0 | -18.0±2.0 | -23.4±2.5 | 0.0±0.0 | 0.0±0.0 | 0.0±0.0 | 0.0±0.0 | -3.2±2.6 | **1.0±0.7** | 0.0±0.0 |
| H2 | 0.0±0.0 | 0.0±0.0 | -5.0±0.3 | 0.0±0.0 | 0.0±0.0 | -1.4±0.2 | 0.0±0.0 | 0.0±0.0 | 0.0±0.0 | -3.6±2.2 | -11.1±1.1 | 0.0±0.0 | 0.0±0.0 | 0.0±0.0 | 0.0±0.0 | 3.5±3.9 | **9.4±7.2** | 0.0±0.0 |
| H3 | 9.8±5.8 | **10.0±6.0** | 5.1±5.9 | 12.0±6.5 | 12.0±6.5 | 10.4±6.5 | 10.0±6.0 | 10.0±6.0 | 9.7±6.0 | -3.5±3.8 | -19.8±4.0 | 0.0±0.0 | 12.0±6.5 | 12.0±6.5 | 10.9±5.9 | 10.9±4.5 | **14.8±6.0** | 0.0±0.1 |
| H4 | 40.0±0.0 | 40.0±0.0 | 35.2±0.4 | 40.0±0.0 | 40.0±0.0 | 38.8±0.1 | 40.0±0.0 | 40.0±0.0 | 40.0±0.0 | -6.3±2.6 | -9.3±1.1 | 0.0±0.0 | 40.0±0.0 | 40.0±0.0 | 38.9±0.4 | 10.8±4.6 | **11.5±1.4** | 0.8±0.7 |
| **Average** | 12.5±11.3 | 12.5±11.3 | 7.7±11.3 | 13.0±11.4 | 13.0±11.4 | 11.6±11.4 | 12.5±11.3 | 12.5±11.3 | 12.4±11.3 | -7.9±4.8 | -15.9±4.6 | 0.0±0.0 | 13.0±11.4 | 13.0±11.4 | 12.4±11.0 | 5.5±5.6 | **9.1±5.8** | 0.2±0.2 |

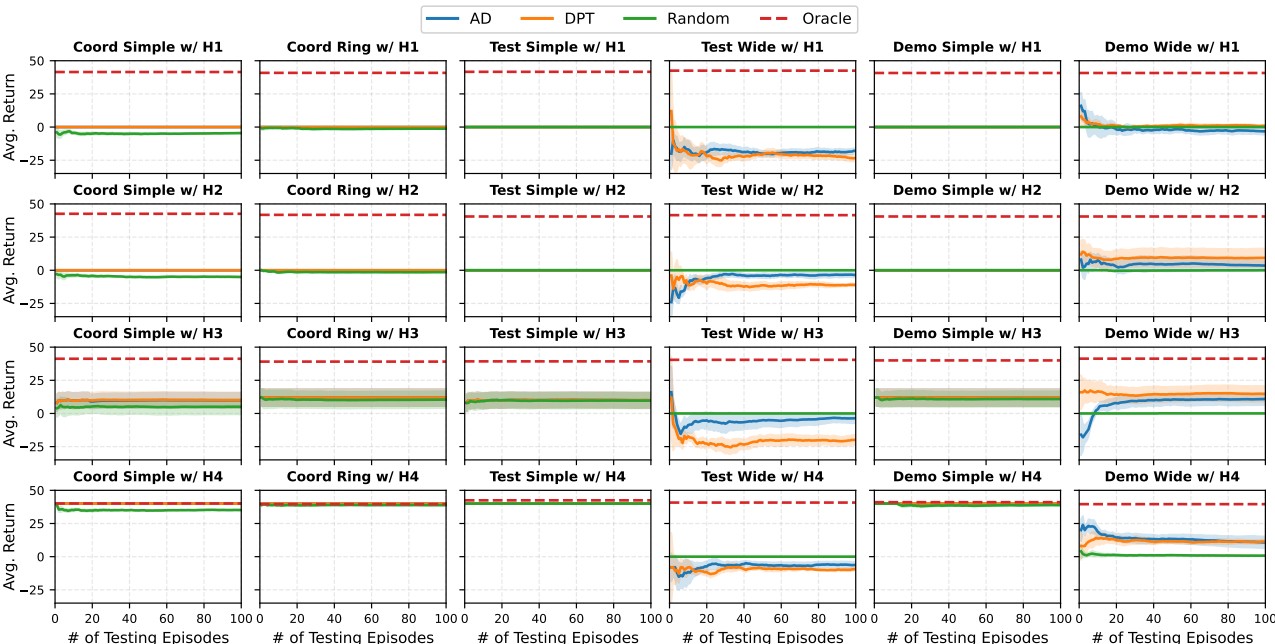

*Figure 3.* **Track 1: Online Adaptation Curves** ($\mathcal{L}_{\text{train}} \times \Pi_{\text{test}}$). Episode-wise return trajectories of AD, DPT, and random baseline across six training layouts and four heuristic teammate families. Unlike single-agent ICRL settings where returns typically increase over the interaction horizon, the studied baselines exhibit ***flat adaptation profiles*** with *no observable in-context improvement*, even as more episodes are accumulated. This visualization complements Table 3 by revealing that the aggregate performance gap stems from a persistent absence of test-time adaptation within these baselines rather than isolated failures on specific episodes.

a per-instance mean return is computed first, and tables report the mean ± standard deviation across 5 such instance-level means. Shaded regions in adaptation curves denote the standard deviation across teammate instances.

**Catastrophic Failure on Complex Coordination.** On the `H1` family, which requires *tight coupling between agents*, AD and DPT frequently yield *near-zero or negative returns*. Notably, in the `test wide` layout with `H1`, AD scores $-18.0 \pm 2.0$ and DPT scores $-23.4 \pm 2.5$, performing **significantly worse than random baseline** $(0.0 \pm 0.0)$. This suggests that the ICRL agents *not only fail to coordinate but actively interfere with the teammate's sub-optimal policy*.

**Absence of In-Context Learning.** A defining characteristic of ICRL is the improvement of policy performance within a single context window as the agent gathers history. Fig. 3 plots the online adaptation curves for AD and DPT. We observe **flat learning curves** across almost all scenarios; episode-wise returns *do not trend upward* over the testing horizon. This stagnation indicates that the underlying Transformer models **fail to perform the necessary Bayesian inference** to identify the teammate's policy $\pi^{-i}$ from the interaction history $h_t^i$. Instead, the agents appear to execute a *fixed, average policy* that is robust only to the training distribution but brittle to out-of-distribution heuristic families. We further quantify within-rollout adaptation via an "adaptation gain" metric (last 20 episodes minus first 20 episodes) in Sec. F.4.4, which confirms the near-absence of online improvement.

### 5.2. Track 2: Layout Generalization

Track 2 evaluates robustness under *severe distribution shifts* by pairing unseen teammates with novel environmental dynamics. We employ layout pairs that share geometric simi-

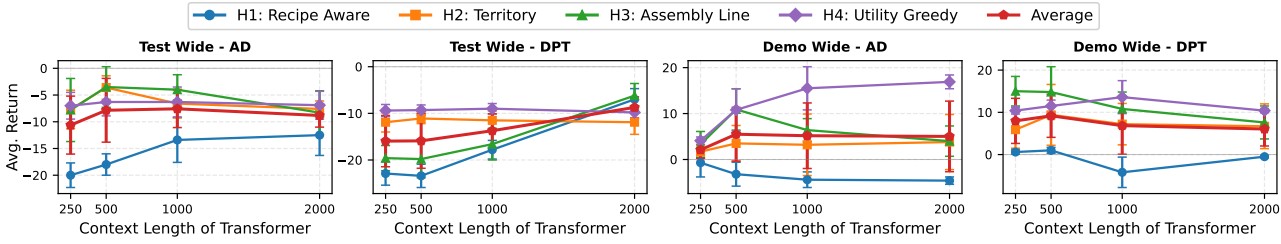

*Figure 4.* **Ablation: Context Length.** Effect of Transformer context length on ICRL performance across two Track 1 layouts (`test wide` and `demo wide`). Each curve represents a different heuristic teammate family. Contrary to expectations from single-agent ICRL, where longer context windows typically enable better in-context adaptation, we observe **no consistent improvement** as context length increases. Performance remains *largely flat or exhibits high variance*, suggesting that current architectures **fail to effectively leverage extended interaction histories** for teammate inference in multi-agent coordination settings.

*Table 4.* **Track 2: Layout Generalization** ($\mathcal{L}_{\text{test}} \times \Pi_{\text{test}}$). Results on held-out layouts with unseen teammates. Agents trained on `asymm adv both` are tested on `asymm adv right`; agents trained on `cramped up` are tested on `cramped down`. Strikingly, **random baseline often matches or exceeds learned ICRL methods**, revealing substantial limitations of the studied baselines in generalizing to novel envs.

| Heuristic Family | asymm adv right | | | cramped down | | |
|---|---|---|---|---|---|---|
| | AD | DPT | Rnd | AD | DPT | Rnd |
| H1 | -2.2±0.9 | **0.0±0.0** | **0.0±0.0** | -0.3±0.4 | **0.0±0.0** | -1.0±1.3 |
| H2 | -0.8±0.6 | **2.7±2.2** | 0.0±0.1 | 0.0±0.0 | 0.0±0.0 | **0.2±0.2** |
| H3 | 2.2±2.5 | **10.3±6.9** | 7.9±5.9 | 0.0±0.0 | 0.0±0.1 | **0.1±0.1** |
| H4 | 18.4±2.5 | **40.0±0.0** | 38.7±0.4 | 3.3±0.9 | 2.3±0.3 | **17.9±0.2** |
| Average | 4.4±5.8 | **13.2±11.2** | 11.7±11.0 | 0.8±1.1 | 0.6±0.7 | **4.3±5.3** |

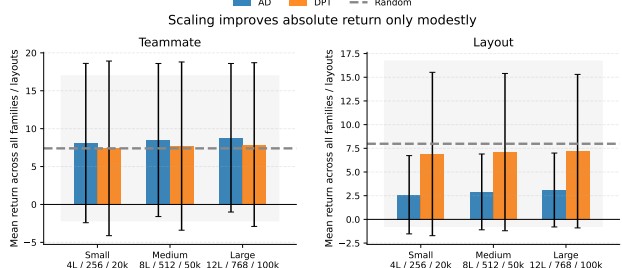

*Figure 5.* **Model Scale and Training Budget Sweep.** Performance of AD and DPT under Small, Medium, and Large configurations across both evaluation tracks. Scaling yields *only modest gains* and **does not restore robust within-context adaptation**.

larities but require **distinct strategic roles**, testing whether agents can re-plan when spatial semantics change. The results are presented in Table 4.

**Structural Generalization Failure.** Across both held-out layouts, ICRL methods **fail to adapt to altered spatial constraints**. Despite geometric similarities to training layouts, successful coordination in test layouts requires *fundamentally different role assignments* and task pipelines. Both AD and DPT produce **near-zero or negative returns**, indicating an inability to infer from interaction history that previously optimal policies are no longer valid in the new environment.

**Random Baseline Dominance.** Strikingly, the **random** (11.7 ± 11.0) **outperforms AD** (4.4 ± 5.8) and remains competitive with DPT (13.2 ± 11.2) on average. This confirms that the studied ICRL baselines *rely heavily on memorizing spatial layout features* during training and **lack the reasoning capabilities to re-plan** in novel environments.

### 5.3. Ablation Studies

We investigate whether the observed failures stem from *insufficient information*, *architectural hyperparameters*, *model capacity*, or *architectural paradigm*.

**Teammate Action Conditioning.** A hypothesis for the fail-

ure of ICRL in AHT is the *partial observability of the partner's actions*. In Table 5, we evaluate variants of AD and DPT conditioned explicitly on Teammate Actions (+TA). The inclusion of teammate actions yields **inconsistent results**. While DPT+TA shows slight improvement on `demo wide` with H2 (9.4 vs 2.8), it degrades performance significantly on `test wide` with H1 (−14.5 vs −23.4, still highly negative). This implies the failure is **not primarily due to missing information** but rather model's *inability to causally link teammate's observed actions to a latent intent*.

**Context Length.** In single-agent meta-RL, longer context windows $K$ typically correlate with better system identification. Fig. 4 illustrates the performance of AD and DPT as the context length increases from $K = 250$ to $K = 2000$ transitions. We observe **no consistent correlation between context length and return**; performance remains flat. This suggests that the "contextual span" required to infer a teammate's policy is either *not being captured by the attention mechanism*, or the model has *failed to learn the meta-exploration policies* required to use this history effectively. Extended context evaluation up to $K = 10,000$ in Sec. F.4.2 confirms that longer histories help somewhat on selected settings but do not qualitatively change the main result.

**Model Scale and Training Budget.** A natural concern with negative results is whether they stem from insufficient model capacity or training budget. We evaluate three configura-

*Table 5.* **Ablation: Teammate Action Conditioning.** Performance comparison of AD and DPT with (+TA) and without teammate action conditioning across challenging layouts. `test wide` and `demo wide` are from Track 1 ($\mathcal{L}_{train} \times \Pi_{test}$), while `cramped down` is a held-out layout from Track 2 ($\mathcal{L}_{test} \times \Pi_{test}$). Adding teammate actions as autoregressive input (+TA) yields *inconsistent improvements* and **fails to address the core limitations** of ICRL methods.

| Heuristic Family | (Track 1) `test wide` | | | | | (Track 1) `demo wide` | | | | | (Track 2) `cramped down` | | | | |
|---|---|---|---|---|---|---|---|---|---|---|---|---|---|---|---|
| | AD | AD+TA | DPT | DPT+TA | Rnd | AD | AD+TA | DPT | DPT+TA | Rnd | AD | AD+TA | DPT | DPT+TA | Rnd |
| H1 | -18.0±2.0 | -2.2±1.0 | -23.4±2.5 | -14.5±2.4 | **0.0±0.0** | -3.2±2.6 | -1.1±1.5 | **1.0±0.7** | 0.7±0.4 | 0.0±0.0 | -0.3±0.4 | -0.0±0.1 | **0.0±0.0** | **0.0±0.0** | -1.0±1.3 |
| H2 | -3.6±2.2 | -5.0±2.8 | -11.1±1.1 | -10.8±1.3 | **0.0±0.0** | 3.5±3.9 | 2.8±1.8 | **9.4±7.2** | 6.8±4.7 | 0.0±0.0 | 0.0±0.0 | 0.0±0.0 | 0.0±0.0 | 0.0±0.0 | **0.2±0.2** |
| H3 | -3.5±3.8 | -1.0±0.9 | -19.8±4.0 | -12.8±2.2 | **0.0±0.0** | 10.9±4.5 | **17.2±4.2** | 14.8±6.0 | 10.9±3.8 | 0.0±0.1 | 0.0±0.0 | 0.0±0.0 | 0.0±0.1 | 0.0±0.0 | **0.1±0.1** |
| H4 | -6.3±2.6 | -6.2±2.2 | -9.3±1.1 | -9.3±1.3 | **0.0±0.0** | 10.8±4.6 | 7.9±2.9 | 11.5±1.4 | **19.1±1.6** | 0.8±0.2 | 3.3±0.9 | 0.6±0.1 | 2.3±0.3 | 5.4±0.7 | **17.9±0.2** |
| Average | -7.9±4.8 | -3.6±2.4 | -15.9±4.6 | -11.8±2.3 | **0.0±0.0** | 5.5±5.6 | 6.7±5.4 | 9.1±5.8 | **9.4±5.5** | 0.2±0.2 | 0.8±1.1 | 0.1±0.2 | 0.6±0.7 | 1.4±1.6 | **4.3±5.3** |

*Table 6.* **Hybrid-AD vs. AD.** Mean return across teammates.

| Track | Layout Setting | AD | Hybrid-AD |
|---|---|---|---|
| Teammate | `test wide` | $-7.9_{\pm 4.8}$ | $-6.5_{\pm 3.8}$ |
| | `demo wide` | $5.5_{\pm 5.6}$ | $6.0_{\pm 5.2}$ |
| Layout | `asymm adv right` | $4.4_{\pm 5.8}$ | $4.7_{\pm 6.0}$ |
| | `cramped down` | $0.8_{\pm 1.1}$ | $0.5_{\pm 1.0}$ |

*Table 7.* **Offline Meta-RL Comparison.** Mean return ($\pm$ std) across all teammate families and layouts on both evaluation tracks. AMAGO-Offline performs comparably to AD/DPT on the Teammate track and *remains weak on the Layout track*.

| Track | AD | DPT | AMAGO-Off. | Rnd |
|---|---|---|---|---|
| Teammate | $8.1_{\pm 10.5}$ | $7.4_{\pm 11.5}$ | **$8.3_{\pm 10.2}$** | $7.4_{\pm 9.6}$ |
| Layout | $2.6_{\pm 4.1}$ | $6.9_{\pm 8.6}$ | $3.4_{\pm 4.8}$ | **$8.0_{\pm 8.7}$** |

tions: **Small** (4 layers, hidden dim 256, 20k steps), **Medium** (8 layers, hidden dim 512, 50k steps), and **Large** (12 layers, hidden dim 768, 100k steps).

As shown in Fig. 5, scaling yields *modest improvements* on Teammate track and *limited gains* on Layout track. The qualitative pattern of **flat learning curves persists** even at the Large configuration, indicating that the failure is **neither paradigm-specific nor capacity-limited**.

**Recurrent Architecture Variant.** To isolate whether *explicit recurrent memory* materially changes the outcome, we evaluate a **Hybrid-AD** variant that replaces the Transformer-based temporal modeling with a CNN+GRU recurrent state tracker while keeping the same AD training protocol. Results are summarized in Table 6.

The hybrid variant yields *marginal improvement* on teammate track, but the gains are *inconsistent* and the Layout track results are essentially unchanged. This confirms that adding explicit recurrent state tracking **does not remove the central difficulty** exposed by the benchmark.

**Additional experimental diagnostics**. We supplement evaluations on held-out RL teammates, extended context, and auxiliary losses in Sec. F.4, which confirm that the observed failures are *robust across a range of diagnostic variations*.

### 5.4. Offline Meta-RL Baseline

Having examined information availability, context length, model capacity, and architectural choice within the AD/DPT paradigm, a natural question is whether a *fundamentally different* training paradigm can overcome the observed fail-

ures. We therefore evaluate an **offline meta-RL baseline** by adopting the AMAGO architecture (Grigsby et al., 2024) and adapting its optimization objective for offline training on the same dataset used by AD and DPT. Unlike AD/DPT, which rely on pure action-prediction, this variant optimizes an *offline RL objective that explicitly maximizes return*.

As shown in Table 7, AMAGO-Offline performs comparably to AD and DPT on the Teammate track and *remains weak on the Layout track*. Crucially, within-context adaptation is *not qualitatively improved*: the offline RL objective does not elicit the rapid partner identification that would manifest as increasing returns over the interaction. This result indicates that the failure exposed by ICRL4AHT is **not specific to the AD/DPT architecture or action-prediction paradigm** but reflects a **deeper challenge**: current sequence-model architectures—regardless of their design or whether they are trained via action-prediction or offline RL—*struggle to perform the implicit Bayesian partner inference* required for AHT under realistic distribution shifts.

## 6. Limitations and Future Work

**Limitations.** Our benchmark has inherent scope constraints: it is instantiated on a *single domain* (OvercookedV2), employs a *finite teammate suite*, and restricts evaluation to *two-player settings with fixed partners*.

**Future Work.** Our findings in this paper motivate developing architectures with *explicit partner inference modules*, training objectives that incentivize *information-seeking exploration*, and extensions to diverse coordination substrates. **A detailed discussion is provided in Sec. G.**

## Acknowledgements

This work is supported in part by the National Key R&D Program of China (No. 2025ZD0122000), the Natural Science Foundation of China (Grant Nos. 62222606 and 6190240), the Key Research and Development Program of Jiangsu Province (Grant No. BE2023016), and the CCF-NetEase ThunderFire Innovation Research Funding (Grant No. 202605).

## Impact Statement

This work introduces ICRL4AHT, a large-scale benchmark with a reproducible evaluation pipeline for in-context reinforcement learning in cooperative multi-agent settings. Our contribution is primarily diagnostic: we systematically characterize the limitations of representative ICRL baselines in ad-hoc teamwork rather than deploying operational systems. The open-source benchmark infrastructure and empirical findings are intended to guide the community toward more robust multi-agent coordination algorithms. We do not foresee direct negative societal consequences, as the work remains foundational in nature. However, downstream applications of improved ad-hoc teamwork methods—such as human-robot collaboration, autonomous vehicle coordination, or multi-agent planning systems—will require careful consideration of safety, reliability, fairness, and alignment with human intent before real-world deployment.

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

# A. OvercookedV2 Environment

This section provides a comprehensive specification of the OvercookedV2 environment used in the ICRL4AHT benchmark. We detail the environment mechanics, our implementation enhancements, and the specific layouts employed across both evaluation tracks.

## A.1. Environment Overview

OvercookedV2 (Gessler et al., 2025) is a cooperative multi-agent environment built upon the original Overcooked-AI (Carroll et al., 2019) framework, introducing critical complexities designed to stress-test coordination algorithms. In this environment, two agents collaborate in a kitchen to prepare and deliver dishes according to dynamically changing recipes under time pressure.

**Game Mechanics.** Each episode proceeds as follows: (1) agents observe the current target recipe displayed on recipe indicators; (2) agents collect ingredients from dispensers (onions, tomatoes, or other ingredient types depending on the layout); (3) agents place ingredients into cooking pots, where each pot requires exactly three ingredients; (4) once filled, pots automatically begin cooking (or require an explicit interaction in certain configurations) for a fixed duration of 20 timesteps; (5) agents collect plates from plate piles and retrieve cooked dishes from pots; (6) agents deliver completed dishes to serving areas. Correct deliveries yield $+20$ reward, while incorrect deliveries incur $-20$ penalty. Upon successful delivery, a new target recipe is sampled from the layout's recipe pool. In layouts with button-activated recipe indicators, agents must interact with the button to reveal the current recipe, incurring a $-5$ cost per activation; the recipe remains visible for 10 timesteps before requiring reactivation.

**Action Space.** Agents select from six discrete actions at each timestep: four directional movements (`up`, `down`, `left`, `right`), a `stay` action, and an `interact` action. The interact action enables agents to pick up items, place items on counters or into pots, start cooking processes, and deliver dishes, depending on the agent's current inventory and the object they are facing.

**Partial Observability.** Unlike the fully observable Overcooked-AI, OvercookedV2 enforces partial observability through a limited $5 \times 5$ field of view centered on each agent. This restriction prevents agents from directly observing distant portions of the kitchen, requiring inference about global state from local observations and partner behavior.

**Stochasticity.** The environment incorporates multiple sources of stochasticity: (1) agent starting positions are randomized at episode initialization within their designated movement areas; (2) target recipes are sampled uniformly from the layout's recipe pool upon reset and after each successful delivery; (3) in layouts with multiple enclosed spaces, agents are constrained to remain within their initial room, introducing systematic variation in coordination requirements.

## A.2. Implementation Details

Our implementation is built entirely in JAX (Bradbury et al., 2018), enabling high-throughput parallel rollouts on GPU/TPU accelerators. We adapted the environment from the JaxMARL (Rutherford et al., 2024) codebase with several bug fixes and optimizations.

**Observation Encoding.** The observation tensor is a three-dimensional array of shape $(5, 5, C)$, where $C$ varies by layout based on the number of available ingredients. The observation channels encode:

- **Agent layers** (per agent): position (1 channel), direction (4 channels one-hot), inventory encoding ($2 + n$ channels where $n$ is the number of ingredient types).
- **Static terrain layers** (6 channels): walls/counters, serving areas, cooking pots, recipe indicators, button-activated recipe indicators, and plate piles.
- **Ingredient dispenser layers** ($n$ channels): one channel per ingredient type indicating dispenser locations.
- **Dynamic object layers** ($2 + n$ channels): plate presence, cooked status, and ingredient counts on counters and in pots.
- **Recipe layers** ($2 + n$ channels): current target recipe encoding displayed on active recipe indicators.
- **Extra layers** (1 channel): pot cooking timer countdown.

- **Delivery indicator layer** (optional, 1 channel): present only in `test simple` and `test wide` layouts to indicate successful delivery events.

The total channel count follows the formula $C = 25 + 5n$ (or $C = 26 + 5n$ for layouts with delivery indicators), where $n$ is the number of ingredient types in the layout.

**Collision Resolution.** When multiple agents attempt to move to the same cell, the environment employs an iterative collision resolution algorithm: (1) identify all position conflicts; (2) revert conflicting agents to their original positions; (3) repeat until no conflicts remain. Additionally, agents are prevented from swapping positions in a single timestep to avoid unrealistic pass-through behavior.

**Recipe Encoding.** Recipes are encoded as bit-packed integers where each ingredient type occupies two bits, allowing counts from 0 to 3 per ingredient. A valid recipe always contains exactly three ingredients total. The cooked status and plate presence are encoded in the two least significant bits, enabling efficient bitwise operations for recipe matching during delivery validation.

### A.3. Track 1 Layouts: Teammate Generalization

Track 1 evaluates adaptation to unseen teammates within familiar environments. We employ six layouts with varying spatial configurations and coordination requirements.

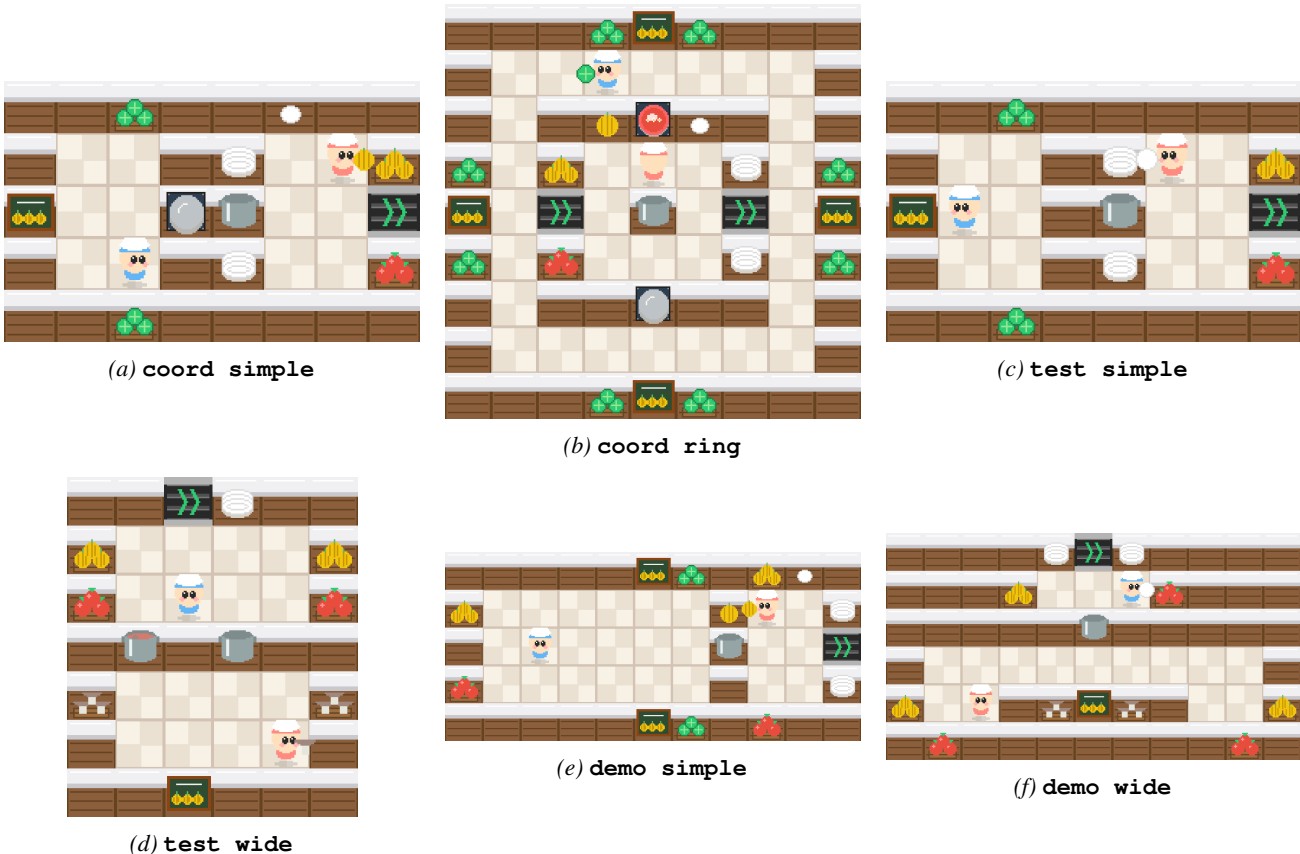

*(a)* `coord simple`

*(b)* `coord ring`

*(c)* `test simple`

*(d)* `test wide`

*(e)* `demo simple`

*(f)* `demo wide`

*Figure 6.* **Track 1 Layouts.** Six layouts used for teammate generalization evaluation. The two agents are rendered as chibi-style chefs with blue (ego) and red (partner) colored bodies; wooden crates with ingredients represent dispensers; metallic pots sit on counters; dark conveyor-belt hatches are serving areas; stacked white plates form plate piles; and chalkboard-style menus or industrial buttons denote recipe indicators.

**`grounded_coord_simple` (coord simple).** This layout (8×5 grid) features a symmetric design with three ingredient types (0, 1, and 2) distributed across the layout, a central pot, and separated serving areas. Agents start in the right region

with access to ingredient types 0 and 1. The layout supports two possible recipes: $[0, 0, 0]$ (three of ingredient 0) and $[1, 1, 1]$ (three of ingredient 1). A button-activated recipe indicator (rendered as an industrial push-button) requires agents to explicitly query the current target, introducing an information acquisition cost. This design tests grounded coordination where agents must agree on which ingredient to prioritize.

**grounded_coord_ring (coord ring).** This larger layout (9×9 grid) extends the grounded coordination problem with a ring topology. Multiple ingredient dispensers and recipe indicators are distributed around the perimeter, with a central structure containing pots and plate piles. Agents can traverse the ring in either direction to access resources. The expanded spatial scale increases the importance of efficient path planning and reduces the frequency of agent collisions.

**test_time_simple (test simple).** This layout (8×5 grid) is designed for test-time protocol formation. Unlike `coord simple`, it removes the button-activated indicator in favor of a standard always-visible recipe indicator. Three ingredient types (0, 1, and 2) are available with symmetric access. The simplified information structure allows focus on behavioral adaptation without the confound of information acquisition strategies.

**test_time_wide (test wide).** This vertically-oriented layout (6×7 grid) presents a challenging configuration with pots at the top, serving area adjacent to the pots, and ingredient dispensers distributed across multiple vertical levels. The layout includes ingredient types 0, 1, and 3 (with multiple dispensers per type), recipe indicators in the lower region, and plate piles near the serving area. The vertical extent and distributed resources create long travel distances and frequent agent interference, making coordination particularly demanding.

**demo_cook_simple (demo simple).** This horizontally-extended layout (11×5 grid) features a long corridor design with the cooking area centrally located and ingredient dispensers at the extremities. Agents must coordinate long-distance ingredient transport while avoiding bottlenecks in the narrow central region. The layout includes button-activated recipe indicators on both sides, requiring explicit information acquisition.

**demo_cook_wide (demo wide).** This layout (11×6 grid) combines horizontal and vertical complexity. The cooking pot is centrally positioned with agents accessing it from above and below. Serving areas and plate piles are at the top, while ingredient dispensers are distributed in the lower regions. Recipe indicators with button activation are positioned in the middle level. This design requires agents to negotiate access to the central pot while managing the multi-stage workflow of ingredient collection, cooking, plating, and delivery.

### A.4. Track 2 Layouts: Layout Generalization

Track 2 evaluates robustness under simultaneous teammate and environment distribution shifts. We employ two training-testing layout pairs that share structural similarities but require distinct coordination strategies.

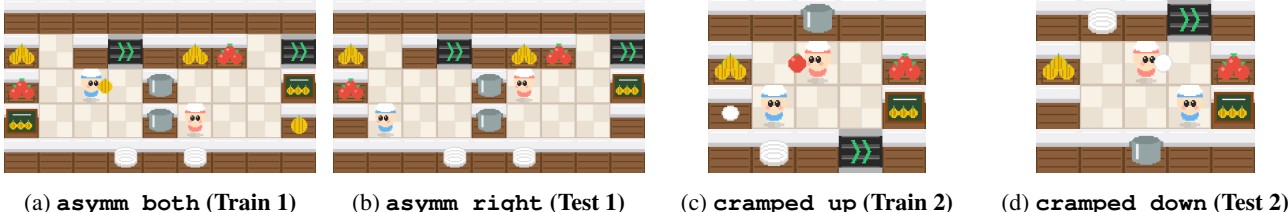

(a) **asymm both (Train 1)**    (b) **asymm right (Test 1)**    (c) **cramped up (Train 2)**    (d) **cramped down (Test 2)**

*Figure 7.* **Track 2 Layout Pairs.** Training and testing layout pairs for layout generalization evaluation. Each pair shares geometric structure but differs in the spatial arrangement of key elements, requiring adaptation of coordination strategies.

**Layout Pair 1: Asymmetric Advantages.** These layouts (9×5 grid) are adapted from the classic Overcooked-AI asymmetric advantages design, featuring two distinct regions separated by a central wall structure with pot access points.

- **asymm_advantages_recipes_both (Train 1):** Recipe indicators are positioned on both the left and right sides of the layout, allowing either agent to observe the current target recipe. Two ingredient types (0 and 1) are available on the left side, with serving areas on both extremities. This symmetric information access enables flexible role assignment during training.

- **asymm_advantages_recipes_right (Test 1):** The recipe indicator is positioned only on the right side, creating information asymmetry. The agent operating in the right region has privileged access to recipe information and must communicate this implicitly through behavior. Successful coordination requires the left-side agent to infer the target recipe from the right-side agent's ingredient selection patterns.

**Layout Pair 2: Cramped Room.**   These compact layouts (5×4 grid) represent the minimal viable cooking environment with high agent density and frequent interaction requirements.

- **cramped_room_up (Train 2):** The cooking pot is positioned at the top of the layout with the serving area at the bottom. Agents start in the middle row with plate piles and recipe indicators adjacent. The top-to-bottom workflow (collect ingredients from sides → cook at top → plate → deliver at bottom) establishes a natural vertical flow.
- **cramped_room_down (Test 2):** This layout inverts the vertical arrangement: the pot is at the bottom and the serving area is at the top. While geometrically a simple reflection, this inversion fundamentally alters the optimal workflow direction. Agents trained on cramped_room_up must adapt their spatial policies to the reversed layout, testing whether ICRL methods can generalize across structural transformations.

### A.5. Layout Configuration Summary

Table 8 summarizes the key configuration parameters for all benchmark layouts.

*Table 8.* **Layout Configuration Summary.** Key parameters for each layout in the ICRL4AHT benchmark. Dimensions are width×height. #Ing. denotes the number of ingredient types; #Recipes denotes the number of possible target recipes; Obs. Channels is the observation tensor depth.

| Layout | Dims | #Ing. | #Recipes | Obs. Ch. | Recipes |
|---|---|---|---|---|---|
| *Track 1: Teammate Generalization* | | | | | |
| coord simple | 8×5 | 3 | 2 | 40 | $[0,0,0], [1,1,1]$ |
| coord ring | 9×9 | 3 | 2 | 40 | $[0,0,0], [1,1,1]$ |
| test simple | 8×5 | 3 | 2 | 41 | $[0,0,0], [1,1,1]$ |
| test wide | 6×7 | 4 | 2 | 46 | $[0,0,0], [1,1,1]$ |
| demo simple | 11×5 | 3 | 2 | 40 | $[0,0,0], [1,1,1]$ |
| demo wide | 11×6 | 4 | 2 | 45 | $[0,0,0], [1,1,1]$ |
| *Track 2: Layout Generalization* | | | | | |
| asymm both (Train 1) | 9×5 | 2 | 2 | 35 | $[0,0,0], [1,1,1]$ |
| asymm right (Test 1) | 9×5 | 2 | 2 | 35 | $[0,0,0], [1,1,1]$ |
| cramped up (Train 2) | 5×4 | 2 | 2 | 35 | $[0,0,0], [1,1,1]$ |
| cramped down (Test 2) | 5×4 | 2 | 2 | 35 | $[0,0,0], [1,1,1]$ |

# B. Teammate Policy Generation

This section details the generation procedures for both RL-trained and heuristic teammate policies used in the ICRL4AHT benchmark. We provide comprehensive descriptions of algorithm principles, hyperparameter configurations, and quality filtering mechanisms to ensure reproducibility.

## B.1. RL Teammate Policies

We employ two primary algorithms for generating diverse RL teammate policies: Fictitious Co-Play (FCP) and Best Response Diversity (BRDiv). Both algorithms build upon Independent Proximal Policy Optimization (IPPO) as the base training procedure.

### B.1.1. INDEPENDENT PPO (IPPO)

IPPO serves as the foundational training algorithm, implementing independent PPO with parameter sharing across agents. Each agent maintains a CNN-RNN policy network that processes grid-based partial observations.

**Network Architecture.** The policy network employs a convolutional neural network followed by a Gated Recurrent Unit (GRU) for temporal reasoning under partial observability. The CNN processes the $(5, 5, C)$ observation tensor, followed by a fully connected layer of dimension 128 and a GRU with hidden dimension 128. The network outputs both action logits and value estimates.

**Training Procedure.** Training proceeds by collecting rollouts of length 256 from 256 parallel environments. Each rollout batch is processed with 4 PPO update epochs using 64 minibatches. We employ Generalized Advantage Estimation (GAE) with $\lambda = 0.95$ and discount factor $\gamma = 0.99$. The policy is optimized using the clipped surrogate objective with $\epsilon_{\text{clip}} = 0.2$, entropy coefficient $0.01$, and value function coefficient $0.5$. Gradients are clipped to maximum norm $0.25$.

**Learning Rate Schedule.** We employ a warmup-then-cosine-decay learning rate schedule. The learning rate linearly increases from 0 to the base rate $(2.5 \times 10^{-4})$ over the first 5% of training updates, then follows cosine decay for the remaining 95%.

**Reward Shaping Annealing.** The environment provides shaped rewards to accelerate early learning. We linearly anneal the reward shaping coefficient from 1.0 to 0.0 over the first $1.5 \times 10^7$ timesteps, ensuring that final policies optimize for the true sparse reward signal:

$$r_{\text{total}} = r_{\text{sparse}} + \alpha(t) \cdot r_{\text{shaped}}, \quad \alpha(t) = \max\left(0, 1 - \frac{t}{1.5 \times 10^7}\right). \tag{3}$$

Table 9 summarizes the complete IPPO hyperparameter configuration.

### B.1.2. FICTITIOUS CO-PLAY (FCP)

FCP (Strouse et al., 2021) generates teammate diversity by training multiple independent IPPO policies in parallel, each initialized with different random seeds. This approach produces a population of policies that converge to different local optima, exhibiting varied behavioral conventions.

**Algorithm Principle.** FCP leverages the observation that RL training with different random initializations naturally produces policies with distinct coordination strategies. By training $N$ independent policies and saving $M$ checkpoints per policy at regular intervals during training, we obtain $N \times M$ teammate candidates spanning different skill levels and behavioral patterns.

**Implementation.** We train $N = 10$ independent IPPO policies using JAX's `vmap` for parallel execution. Each policy saves $M = 5$ checkpoints at evenly-spaced intervals throughout training, yielding 50 candidate teammates per layout. The checkpoints capture policies at different stages of learning, from early exploration to near-convergence, providing behavioral diversity along the skill dimension.

*Table 9.* **IPPO Hyperparameters.** Complete configuration for Independent PPO training used as the base algorithm for all RL teammate generation.

| Parameter | Value | Description |
|---|---|---|
| *Training Configuration* | | |
| Total timesteps | $3 \times 10^7$ | Total environment steps per policy |
| Parallel environments | 256 | Number of vectorized environments |
| Rollout length | 256 | Steps collected per rollout |
| Update epochs | 4 | PPO epochs per rollout batch |
| Minibatches | 64 | Number of minibatches per epoch |
| *PPO Hyperparameters* | | |
| Learning rate | $2.5 \times 10^{-4}$ | Base learning rate |
| LR warmup fraction | 0.05 | Fraction of updates for linear warmup |
| Discount factor ($\gamma$) | 0.99 | Reward discounting |
| GAE lambda ($\lambda$) | 0.95 | GAE bias-variance tradeoff |
| Clip epsilon | 0.2 | PPO clipping threshold |
| Entropy coefficient | 0.01 | Entropy regularization weight |
| Value function coefficient | 0.5 | Value loss weight |
| Max gradient norm | 0.25 | Gradient clipping threshold |
| *Network Architecture* | | |
| Architecture type | CNN-RNN | Convolutional + recurrent |
| FC dimension | 128 | Fully connected layer size |
| GRU hidden dimension | 128 | Recurrent hidden state size |
| Activation | ReLU | Nonlinearity function |
| *Environment* | | |
| Episode length | 400 | Maximum timesteps per episode |
| Reward shaping horizon | $1.5 \times 10^7$ | Steps to anneal shaping to zero |

### B.1.3. BEST RESPONSE DIVERSITY (BRDIV)

BRDiv (Rahman et al., 2023) generates diverse teammates through an adversarial training objective that explicitly optimizes for behavioral distinctiveness.

**Algorithm Principle.** BRDiv maintains two populations: confederate policies and their corresponding best responses. Confederate policies are trained to achieve high self-play returns while minimizing cross-play returns when paired with best responses trained against other confederates. This objective encourages confederates to develop mutually incompatible coordination strategies.

**Training Objective.** Let $\pi_k$ denote the $k$-th confederate policy and $\mathrm{BR}_k$ its best response. The confederate loss combines self-play (SP) and cross-play (XP) terms:

$$\mathcal{L}_{\mathrm{conf}}(\pi_k) = -w_{\mathrm{SP}} \cdot J_{\mathrm{SP}}(\pi_k, \pi_k) + w_{\mathrm{XP}} \cdot \sum_{j \neq k} J_{\mathrm{XP}}(\pi_k, \mathrm{BR}_j), \tag{4}$$

where $J_{\mathrm{SP}}$ and $J_{\mathrm{XP}}$ denote expected returns under self-play and cross-play respectively. The weighting scheme uses $w_{\mathrm{SP}} = 1 + 2 \cdot w_{\mathrm{XP}}$ with $w_{\mathrm{XP}} = 0.5$ as the cross-play loss weight parameter. Best response policies are trained to maximize returns when paired with their assigned confederate.

**Implementation.** We train populations of size 10 with 5 checkpoints per policy. Cross-play evaluation uses 20 episodes of maximum length 400 steps to estimate $J_{\mathrm{XP}}$.

### B.1.4. LAGRANGIAN BEST RESPONSE DIVERSITY (LBRDIV)

LBRDiv (Rahman et al., 2024) extends BRDiv by formulating diversity generation as a constrained optimization problem solved via Lagrangian relaxation, enabling more principled control over the diversity-performance tradeoff.

**Algorithm Principle.** LBRDiv reformulates the diversity objective as a set of pairwise constraints ensuring that each confederate policy achieves higher self-play returns than cross-play returns with other confederates' best responses, subject to a tolerance margin. Specifically, for confederate $i$ with best response $\text{BR}_i$, the constraints require:

$$J(\pi_i, \text{BR}_i) \geq J(\pi_j, \text{BR}_i) + \epsilon, \quad \forall j \neq i, \tag{5}$$

where $\epsilon$ is a tolerance factor ensuring meaningful performance gaps. This formulation guarantees that each confederate-BR pair achieves distinctly higher returns than any mismatched pairing.

**Lagrangian Dual Formulation.** LBRDiv maintains two Lagrange multiplier matrices: $\Lambda^{\text{vert}}$ enforcing that confederate $i$ outperforms confederate $j$ when paired with $\text{BR}_i$, and $\Lambda^{\text{horiz}}$ enforcing that confederate $i$ paired with $\text{BR}_i$ outperforms confederate $i$ paired with $\text{BR}_j$. The policy gradient loss weights are derived from these multipliers, automatically balancing self-play maximization against cross-play minimization. The Lagrange multipliers are updated via gradient descent on the constraint violations:

$$\Lambda_{ij}^{\text{vert}} \leftarrow \max\left(\Lambda_{ij}^{\text{vert}} - \eta_\Lambda \cdot \left(J(\pi_i, \text{BR}_i) - J(\pi_j, \text{BR}_i) - \epsilon\right), 0.5 \cdot \mathbb{1}_{i=j}\right), \tag{6}$$

where $\eta_\Lambda$ is the Lagrange multiplier learning rate. Diagonal elements are fixed at $0.5$ to maintain self-play optimization weight.

**Implementation.** We adopt the same population structure as BRDiv with 10 confederate-BR pairs and 5 checkpoints per policy. Table 10 summarizes the LBRDiv-specific hyperparameters.

*Table 10.* **LBRDiv-Specific Hyperparameters.** Additional hyperparameters beyond the base IPPO configuration for Lagrangian Best Response Diversity training.

| Parameter | Value | Description |
| --- | --- | --- |
| Population size | 10 | Number of confederate-BR pairs |
| Checkpoints per policy | 5 | Saved checkpoints during training |
| Tolerance factor ($\epsilon$) | 0.1 | Minimum performance gap for constraints |
| Lagrange LR ($\eta_\Lambda$) | 0.01 | Learning rate for multiplier updates |
| Evaluation episodes | 20 | Episodes for cross-play return estimation |
| Evaluation max steps | 400 | Maximum steps per evaluation episode |

### B.1.5. COOPERATIVE MIXED-PLAY DIVERSITY (COMEDI)

CoMeDi (Sarkar et al., 2024) generates diverse teammates through an iterative population expansion procedure that explicitly optimizes for complementary coordination strategies via mixed-play interactions.

**Algorithm Principle.** Unlike simultaneous population training in BRDiv/LBRDiv, CoMeDi grows the population incrementally. Starting from a single IPPO-trained policy, each iteration adds a new confederate policy trained to: (1) achieve high self-play returns with its own best response, (2) achieve *low* cross-play returns when paired with existing population members (encouraging incompatibility), and (3) perform well under mixed-play conditions where the partner randomly switches between the ego policy and the best response.

**Training Objective.** For a newly trained confederate $\pi_{\text{new}}$ with best response $\text{BR}_{\text{new}}$, and an existing population member $\pi_{\text{ego}}$ selected as the most compatible partner, the CoMeDi objective combines three terms:

$$\mathcal{L}_{\text{CoMeDi}} = -J_{\text{SP}}(\pi_{\text{new}}, \text{BR}_{\text{new}}) + \alpha \cdot J_{\text{XP}}(\pi_{\text{new}}, \pi_{\text{ego}}) - \beta \cdot J_{\text{MP}}(\pi_{\text{new}}, \text{mix}), \tag{7}$$

where $J_{\text{MP}}$ denotes expected returns under mixed-play (partner randomly samples from $\{\pi_{\text{ego}}, \text{BR}_{\text{new}}\}$ at each episode), $\alpha$ is the cross-play loss weight encouraging incompatibility with existing policies, and $\beta$ is the mixed-play loss weight encouraging robustness to partner uncertainty.

**Partner Selection.** At each iteration, CoMeDi evaluates the new policy against all existing population members and selects the partner yielding the highest cross-play returns as the $\pi_{\text{ego}}$ for adversarial training. This adaptive selection ensures that training focuses on differentiating from the most similar existing policy.

**Implementation.** We initialize with one IPPO policy and iteratively add 9 confederates for a final population of 10. Each confederate training uses 4 parallel rollout types: self-play (SP), cross-play with selected ego (XP), mixed-play for state distribution (MP), and mixed-play starting from MP-visited states (SMP). Table 11 summarizes the CoMeDi-specific hyperparameters.

*Table 11.* **CoMeDi-Specific Hyperparameters.** Additional hyperparameters beyond the base IPPO configuration for Cooperative Mixed-Play Diversity training.

| Parameter | Value | Description |
|---|---|---|
| Population size | 10 | Final number of confederate policies |
| Checkpoints per policy | 5 | Saved checkpoints during training |
| Cross-play weight ($\alpha$) | 1.0 | Weight for XP loss (encourages incompatibility) |
| Mixed-play weight ($\beta$) | 1.0 | Weight for MP loss (encourages robustness) |
| Timesteps per iteration | $3 \times 10^7$ | Training steps for each new confederate |
| Argmax rollout episodes | 20 | Episodes for partner selection evaluation |

### B.1.6. CHECKPOINT QUALITY FILTERING

To ensure teammate quality, we implement a filtering pipeline that selects high-performing checkpoints from the candidate pool.

**Filtering Procedure.** For each layout, the filtering process proceeds as follows:

1. **Discovery**: Locate all checkpoint files across training runs for the target layout.

2. **Return Evaluation**: Load the mean episode return recorded for each checkpoint during training.

3. **Threshold Filtering**: Retain only checkpoints with mean return exceeding the minimum threshold (default: 20).

4. **Random Sampling**: From the filtered set, uniformly sample the target number of teammates (default: 20) using a fixed random seed for reproducibility.

This procedure reduces the initial 50 candidates to 20 high-quality teammates, ensuring that the training distribution consists of competent partners capable of meaningful coordination.

### B.2. Heuristic Teammate Policies

We design four heuristic policy families that implement distinct coordination archetypes with varying levels of autonomous task-completion capability. Each family is parameterized by a set of continuous and discrete hyperparameters that control behavioral nuances, enabling systematic coverage of the coordination difficulty spectrum.

### B.2.1. COMMON INFRASTRUCTURE

All heuristic agents share foundational components:

**Pathfinding.** Agents employ BFS-based precomputed distance matrices and next-action matrices at initialization. Given the static layout structure, these matrices enable $O(1)$ path queries during execution.

**Parameterization.** Each family defines a parameter dataclass (`Theta`) containing tunable hyperparameters. Parameters support both discrete presets and continuous sampling for generating behavioral variants.

### B.2.2. H1: RECIPE-AWARE AGENT

The Recipe-Aware agent implements interpretable decision logic centered on recipe knowledge acquisition and adherence.

**Core Principle.** This agent maintains an internal belief about the current recipe and uses the button-activated recipe indicator ('L') to query the recipe when unknown or stale. The decision priority follows: (1) deliver held dishes, (2) plate cooked soups, (3) add ingredients to pots matching the known recipe, and (4) acquire recipe information when uncertain.

**Key Parameters.** Table 12 summarizes the Recipe-Aware agent parameters.

*Table 12.* **H1: Recipe-Aware Agent Parameters.** Hyperparameters controlling recipe knowledge acquisition and adherence behavior.

| Parameter | Range | Description |
|---|---|---|
| press_L_when_unknown | $[0, 1]$ | Probability of pressing L when recipe unknown |
| refresh_interval | $[50, 200]$ | Steps before considering recipe stale |
| strict_recipe | $[0, 1]$ | Adherence to recipe (1=strict, 0=default ingredient) |
| plate_timing | $[0, 1]$ | Plate fetching timing (1=early, 0=wait until cooked) |
| dist_weight | $[0.1, 1.0]$ | Distance penalty weight |
| inertia | $[0, 1]$ | Bonus for maintaining current intent |
| idle_prob | $[0, 1]$ | Probability of staying idle each step |
| wrong_ingredient_prob | $[0, 1]$ | Probability of selecting wrong ingredient |

### B.2.3. H2: TERRITORY AGENT

The Territory agent implements region-constrained behavior where each agent operates primarily within a designated spatial territory.

**Core Principle.** The kitchen is partitioned into territories based on configurable split modes: vertical (left/right), horizontal (top/bottom), or functional (prep/service stations). Agents are biased to act within their assigned territory, with configurable strictness and rescue thresholds for urgent cross-territory interventions.

**Split Modes.**

- **Vertical**: Agent 0 operates on the left side; Agent 1 on the right.
- **Horizontal**: Agent 0 operates on the top; Agent 1 on the bottom.
- **Object Stations**: Agent 0 handles prep (pots, ingredients); Agent 1 handles service (goals, plates).

**Behavior Modes.** The agent supports multiple behavior modes that modulate cooperation difficulty:

- **Normal**: Standard cooperative behavior within territory constraints.
- **Blocker**: Positions in chokepoints with high stay probability (70%).
- **Hoarder**: Picks up items but drops them on counters without task completion.
- **Lazy**: Rarely acts, primarily stays stationary.
- **Invader**: Operates in partner's territory, causing collisions.

**Key Parameters.** Table 13 summarizes the Territory agent parameters.

*Table 13.* **H2: Territory Agent Parameters.** Hyperparameters controlling spatial territory assignment and boundary behavior.

| Parameter | Range | Description |
|---|---|---|
| split_mode | $\{0, 1, 2\}$ | Territory division: vertical/horizontal/functional |
| strictness | $[0, 1]$ | Penalty for acting outside territory |
| shared_margin | $[0, 3]$ | Rows/columns treated as shared corridor |
| rescue_threshold | $[0, 1]$ | Urgency threshold for cross-territory action |
| yield_bias | $[0, 1]$ | Tendency to yield to avoid deadlocks |
| behavior_mode | $\{0, \dots, 5\}$ | Behavior strategy (normal to invader) |
| action_probability | $[0, 1]$ | Action probability for lazy mode |

### B.2.4. H3: ASSEMBLY LINE AGENT

The Assembly Line agent implements role-based cooperative behavior mimicking factory workflow specialization.

**Core Principle.** Agents specialize in distinct workflow stages: ingredient running (fetching and adding ingredients to pots) or plating/delivery (fetching plates, collecting cooked dishes, delivering). This explicit role division creates predictable, structured cooperation patterns.

**Role Modes.**

- **Ingredient Runner**: Exclusively fetches ingredients and adds them to pots.

- **Plater/Deliverer**: Exclusively handles plates, cooked dish collection, and delivery.

- **Flexible**: Dynamically switches roles based on pot states (cooking/cooked triggers plater behavior).

**Handoff Styles.** Agents stage items for partner handoff using configurable strategies:

- **Pot-Adjacent**: Stage items on counters adjacent to cooking pots.

- **Central**: Stage items near the center of the walkable area.

- **Teammate-Nearby**: Stage items near the partner's current position.

**Key Parameters.** Table 14 summarizes the Assembly Line agent parameters.

*Table 14.* **H3: Assembly Line Agent Parameters.** Hyperparameters controlling role assignment and workflow coordination.

| Parameter | Range | Description |
|---|---|---|
| role_mode | $\{0, 1, 2\}$ | Role: runner/plater/flexible |
| handoff_style | $\{0, 1, 2\}$ | Staging location strategy |
| plate_urgency | $[0, 1]$ | Timing for plate fetching |
| prestage_bias | $[0, 1]$ | Tendency to stage extra ingredients |
| start_cook_bias | $[0, 1]$ | Priority for starting cooking |
| hesitation_prob | $[0, 1]$ | Probability of pausing instead of acting |
| wrong_action_prob | $[0, 1]$ | Probability of taking random wrong action |
| task_abandon_prob | $[0, 1]$ | Probability of abandoning current task |

### B.2.5. H4: UTILITY GREEDY AGENT

The Utility Greedy agent implements a rational opportunist that selects actions by maximizing a weighted utility function over candidate intents.

**Core Principle.** At each timestep, the agent enumerates a fixed set of 8 candidate intents, computes utility scores based on weighted value estimates and distance costs, and executes the highest-scoring intent. This approach enables flexible, situation-adaptive behavior without explicit role assignment.

**Intent Types.** The agent considers 8 micro-goals:

1. **Deliver Dish**: Deliver held dish to serving area.

2. **Pickup Cooked**: Collect cooked soup with plate.

3. **Get Plate**: Fetch a plate from pile.

4. **Add Ingredient**: Add held ingredient to pot.

5. **Fetch Ingredient**: Retrieve ingredient from dispenser.

6. **Stage on Counter**: Place item on counter for handoff.

7. **Start Cooking**: Initiate cooking for a full pot.

8. **Press L**: Query recipe via button indicator.

**Utility Scoring.** Each intent receives a score computed as:

$$\text{score}(i) = w_i \cdot v_i - w_{\text{dist}} \cdot d_i + \mathbb{1}[i = i_{\text{prev}}] \cdot w_{\text{inertia}}, \tag{8}$$

where $w_i$ is the intent weight, $v_i$ is a state-dependent value multiplier, $d_i$ is the distance to the target, and $w_{\text{inertia}}$ provides continuity by favoring the previous intent.

**Key Parameters.** Table 15 summarizes the Utility Greedy agent parameters.

*Table 15.* **H4: Utility Greedy Agent Parameters.** Hyperparameters controlling utility weights for intent selection.

| Parameter | Range | Default | Description |
|---|---|---|---|
| w_deliver | $[5, 20]$ | 10.0 | Weight for delivering dish |
| w_pickup_cooked | $[3, 15]$ | 8.0 | Weight for picking up cooked soup |
| w_get_plate | $[2, 10]$ | 5.0 | Weight for getting a plate |
| w_add_ingredient | $[3, 12]$ | 6.0 | Weight for adding ingredient |
| w_fetch_ingredient | $[2, 10]$ | 4.0 | Weight for fetching ingredient |
| w_stage_on_counter | $[0.5, 5]$ | 2.0 | Weight for counter staging |
| w_start_cooking | $[3, 15]$ | 7.0 | Weight for starting cooking |
| w_press_L | $[1, 8]$ | 3.0 | Weight for pressing L button |
| dist_weight | $[0.1, 1.5]$ | 0.5 | Distance penalty coefficient |
| inertia | $[0, 1]$ | 0.3 | Bonus for maintaining intent |

## B.3. Parameter Sampling for Train/Test Splits

To enable controlled distribution shift experiments, we implement a parameter sampling scheme that generates disjoint parameter ranges for training and testing splits.

**Sampling Procedure.** For each heuristic family, we define train and test parameter ranges such that certain dimensions are held out. For example, the Assembly Line agent uses role_mode $\in \{0, 2\}$ (runner, flexible) during training and role_mode $= 1$ (plater) during testing. Similarly, continuous parameters like plate_urgency are partitioned into non-overlapping ranges ($[0, 0.6]$ for train, $[0.6, 1.0]$ for test).

**Reproducibility.** Parameter sampling employs SHA256-based deterministic PRNG seeding from specification strings, ensuring reproducible teammate generation across different compute environments.

# C. RL Source Algorithm Training

This section details the training procedure for the reinforcement learning algorithms used to generate learning histories in the ICRL4AHT dataset. Each learning history captures the complete interaction trajectory of an ego agent learning to coordinate with a fixed teammate policy from scratch.

## C.1. Training Framework Overview

The data collection pipeline employs a task-parallel architecture where each *task* corresponds to training a fresh ego agent against a specific teammate policy on a designated layout. The pipeline consists of two primary components:

1. **Task Runner**: Executes single-task training using PPO with full history recording.

2. **Batch Collector**: Orchestrates parallel execution across all tasks in a manifest, with checkpoint-based resume semantics.

## C.2. PPO Training Procedure

We employ Proximal Policy Optimization (PPO) as the ego agent training algorithm, chosen for its stable learning dynamics and widespread adoption in cooperative multi-agent settings.

### C.2.1. NETWORK ARCHITECTURE

The ego policy employs a CNN-RNN architecture identical to the teammate policies described in Sec. B:

**Convolutional Encoder.** The partial observation tensor of shape $(5, 5, C)$ is processed by a convolutional neural network that extracts spatial features from the agent's local field of view.

**Recurrent Core.** A Gated Recurrent Unit (GRU) with hidden dimension 128 processes the encoded observations sequentially, enabling the agent to maintain memory of past interactions and infer teammate behavior patterns over time.

**Output Heads.** The network produces both action logits (passed through a categorical distribution with invalid action masking) and value estimates for the critic.

### C.2.2. TRAINING CONFIGURATION

Training proceeds by collecting rollouts from vectorized environments and performing batched PPO updates. Table 16 summarizes the complete hyperparameter configuration.

### C.2.3. LEARNING RATE SCHEDULE

We employ a warmup-then-cosine-decay learning rate schedule to ensure stable early training and graceful convergence:

$$\eta(t) = \begin{cases} \eta_{\text{base}} \cdot \frac{t}{t_{\text{warmup}}} & \text{if } t < t_{\text{warmup}} \\ \eta_{\text{base}} \cdot \frac{1}{2} \left( 1 + \cos \left( \pi \cdot \frac{t - t_{\text{warmup}}}{T - t_{\text{warmup}}} \right) \right) & \text{otherwise} \end{cases} \quad (9)$$

where $\eta_{\text{base}} = 2.5 \times 10^{-4}$ is the peak learning rate, $t_{\text{warmup}} = 0.1 \cdot T$ is the warmup period (10% of total updates), and $T$ is the total number of update steps.

### C.2.4. REWARD SHAPING ANNEALING

The Overcooked environment provides intermediate shaped rewards (e.g., for picking up ingredients, adding items to pots) that accelerate early learning. To ensure policies ultimately optimize for the true sparse reward signal (successful dish delivery), we linearly anneal the shaped reward coefficient:

$$r_{\text{train}}(t) = r_{\text{sparse}} + \alpha(t) \cdot r_{\text{shaped}}, \quad \alpha(t) = \max \left( 0, 1 - \frac{t}{H_{\text{shape}}} \right), \quad (10)$$

where $H_{\text{shape}} = 1.5 \times 10^7$ timesteps. This schedule provides full shaping guidance during early exploration and transitions to pure sparse rewards by the end of training.

*Table 16.* **Ego PPO Training Hyperparameters.** Complete configuration for training ego agents that generate learning histories in the ICRL4AHT dataset.

| Parameter | Value | Description |
|---|---|---|
| *Training Configuration* | | |
| Total timesteps | $6 \times 10^7$ | Total environment steps per learning history |
| Parallel environments | 1024 | Number of vectorized environments |
| Rollout length | 256 | Steps collected per rollout before update |
| Update epochs | 4 | PPO epochs per rollout batch |
| Minibatches | 64 | Number of minibatches per epoch |
| *PPO Hyperparameters* | | |
| Learning rate | $2.5 \times 10^{-4}$ | Base learning rate |
| LR warmup fraction | 0.1 | Fraction of updates for linear warmup |
| Discount factor ($\gamma$) | 0.99 | Reward discounting |
| GAE lambda ($\lambda$) | 0.95 | GAE bias-variance tradeoff |
| Clip epsilon | 0.2 | PPO clipping threshold |
| Entropy coefficient | 0.01 | Entropy regularization weight |
| Value function coefficient | 0.5 | Value loss weight |
| Max gradient norm | 0.25 | Gradient clipping threshold |
| *Network Architecture* | | |
| Architecture type | CNN-RNN | Convolutional encoder + GRU |
| FC dimension | 128 | Fully connected layer size |
| GRU hidden dimension | 128 | Recurrent hidden state size |
| Activation | ReLU | Nonlinearity function |
| *Environment & Recording* | | |
| Episode length | 400 | Maximum timesteps per episode |
| Reward shaping horizon | $6 \times 10^7$ | Steps to anneal shaping to zero |
| Recorded environments | 1024 | Number of parallel envs recorded |
| Recorded steps per episode | 100 | First $N$ steps recorded per episode |

## C.3. History Recording

During training, we record complete interaction trajectories to construct learning histories for the ICRL dataset. The recording process operates continuously throughout training with the following specifications:

**Parallel Recording.** All 1024 parallel environments are recorded simultaneously, capturing diverse interaction experiences from the same training run.

**Temporal Truncation.** To manage storage requirements while preserving the most informative learning signal, we record only the first 100 timesteps of each episode. This captures the critical early-episode coordination phase where teammate behavior identification is most important.

**Recorded Data.** For each timestep, we store:

- Ego agent observation (partial grid view)

- Ego agent action taken

- Teammate action taken (for supervision in imitation-based methods)

- Base reward received (sparse dish delivery reward)

- Episode termination signal

**Incremental Saving.** To prevent memory exhaustion during long training runs, history data is saved incrementally to disk in compressed chunks at regular intervals (every 10 updates by default).

## C.4. Batch Execution Pipeline

The batch collector orchestrates parallel training across all tasks defined in a JSONL manifest file.

**Task Manifest.**   Each entry in the manifest specifies: task identifier, layout name, teammate specification (heuristic family with parameters or RL checkpoint path), data split assignment, and random seed.

**Idempotent Execution.**   The pipeline implements checkpoint-based resume semantics. A task is considered complete if and only if valid `history.npz`, `episodes.json`, and `metadata.json` artifacts exist in the output directory. Complete tasks are automatically skipped on re-execution.

**Parallelization.**   Multiple tasks can execute concurrently using process-level parallelism. Lock files prevent duplicate execution when multiple workers process the same manifest. GPU resources are distributed across workers in round-robin fashion when multiple GPUs are available.

**Failure Handling.**   Failed tasks are logged to a structured JSONL file with full stack traces. The pipeline supports configurable retry attempts and fail-fast mode for debugging.

# D. Dataset Structure

This section provides a comprehensive specification of the ICRL4AHT dataset structure, enabling reproducibility and facilitating the development of novel ICRL algorithms on our benchmark. The dataset is serialized in HDF5 format with an accompanying JSONL index for efficient metadata queries.

## D.1. HDF5 Storage Format

Each learning history is stored as an independent HDF5 group, indexed by a sequential integer identifier. The storage employs gzip compression (level 6) with chunk sizes optimized for time-axis slicing during context window sampling. Table 17 enumerates all data fields stored per learning history.

*Table 17.* **Dataset Fields.** Complete specification of data fields stored per learning history in the HDF5 dataset. Shape notation: $T$ denotes total timesteps in the history (default: 14,600); $(H, W, C)$ denotes the spatial observation dimensions where $H = W = 5$ (field of view) and $C$ varies by layout.

| Field Name | Data Type | Shape | Description |
|---|---|---|---|
| obs | float32 | $(T, H, W, C)$ | Ego agent partial observation. Each timestep contains a $5 \times 5$ spatial grid with $C$ channels encoding terrain, objects, agent positions, and recipe state. |
| actions | int32 | $(T,)$ | Ego agent discrete actions. Values in $\{0, 1, 2, 3, 4, 5\}$ correspond to {up, down, left, right, stay, interact}. |
| rewards | float32 | $(T,)$ | Scalar rewards received by the team. Positive for correct deliveries, negative for incorrect deliveries. |
| dones | bool | $(T,)$ | Episode termination flags. `True` indicates the final timestep of an episode within the history. |
| teammate_actions | int32 | $(T,)$ | Partner agent actions (optional). Same action space as ego agent. Enables teammate-action-conditioned variants (+TA). |
| expert_actions | int32 | $(T,)$ | Expert actions from the converged policy (optional). Generated via relabeling using final checkpoint. Used for DPT-style supervised training. |

## D.2. Observation Channels

The observation tensor encodes the ego agent's partial view of the environment state. The channel dimension $C$ varies by layout due to differences in available ingredients and recipe complexity. Table 18 details the observation channel semantics.

## D.3. Index File Structure

The companion JSONL index file enables $O(1)$ metadata lookup without loading the full HDF5 file. Each line contains a JSON object with the fields specified in Table 19.

## D.4. Batch Formats for ICRL Training

The data loader constructs batches tailored to different ICRL paradigms. Table 20 and Table 21 specify the batch formats for AD and DPT training, respectively.

*Table 18.* **Observation Channel Semantics.** The observation channels encode spatial information within the $5 \times 5$ field of view. Channel count $C$ varies by layout: 40 for **coord simple**, **coord ring**, and **demo simple**; 41 for **test simple**; 45 for **demo wide**; 46 for **test wide**; and 35 for Track 2 layouts (**asymm both**, **asymm right**, **cramped up**, **cramped down**).

| Channel Category | Encoding | Description |
|---|---|---|
| Terrain Features | Binary | Spatial encoding of walls, counters, cooking stations, serving areas, and ingredient sources (onion/tomato dispensers). |
| Object States | Binary/Scalar | Presence and state of objects on counters and in cooking pots, including ingredient counts and cooking progress. |
| Agent Positions | Binary | One-hot spatial encoding of ego agent and partner agent locations within the field of view. |
| Held Items | Binary | Encoding of items currently held by each agent (raw ingredients, cooked dishes, plates). |
| Recipe Information | Binary | Current target recipe specification indicating required ingredients and quantities. |

*Table 19.* **Index Entry Fields.** Metadata fields stored per learning history in the JSONL index file, enabling efficient filtering and sampling during training.

| Field Name | Data Type | Description |
|---|---|---|
| history_id | int | Unique sequential identifier for the learning history. |
| task_id | str | Task identifier encoding layout and teammate configuration. |
| env_idx | int | Environment stream index within the parallel rollout. |
| T | int | Total number of timesteps in the history. |
| obs_shape | list[int] | Observation shape as $[H, W, C]$, *e.g.*, $[5, 5, 40]$. |
| action_dim | int | Action space dimensionality (always 6 for OvercookedV2). |
| track | str | Evaluation track: teammate or layout. |
| split | str | Data split: train or test. |
| layout | str | Layout name, *e.g.*, **coord_simple**, **test_wide**. |
| teammate_family | str | Teammate policy family: rl or heuristic families H1–H4. |
| teammate_kind | str | Specific teammate variant within the family. |
| h5_group | str | HDF5 group path for data access, *e.g.*, /0, /1. |
| has_teammate_actions | bool | Whether teammate_actions field is available. |
| has_expert_actions | bool | Whether expert_actions field is available. |

*Table 20.* **AD Batch Format.** Batch structure for Algorithm Distillation training. Notation: $B$ = batch size, $L$ = sequence length (context window), $(H, W, C)$ = spatial observation dimensions.

| Field Name | Data Type | Shape | Description |
|---|---|---|---|
| obs | float32 | $(B, L, H, W, C)$ | Spatial observations for sequence modeling. |
| prev_actions | int32 | $(B, L)$ | Previous timestep actions (0 for $t = 0$). |
| prev_rewards | float32 | $(B, L)$ | Previous timestep rewards (0 for $t = 0$). |
| target_actions | int32 | $(B, L)$ | Actions to predict (training targets). |
| dones | bool | $(B, L)$ | Episode termination flags. |
| attention_mask | bool | $(B, L)$ | Valid position mask (1=valid, 0=padding). |
| prev_teammate_actions | int32 | $(B, L)$ | Previous teammate actions (optional, for +TA). |

*Table 21.* **DPT Batch Format.** Batch structure for Decision-Pretrained Transformer training. Notation: $B$ = batch size, $K$ = context length, $(H, W, C)$ = spatial observation dimensions.

| Field Name | Data Type | Shape | Description |
|---|---|---|---|
| query_obs | float32 | $(B, H, W, C)$ | Query observation for action prediction. |
| query_target | int32 | $(B, )$ | Expert action (supervision target). |
| context_obs | float32 | $(B, K, H, W, C)$ | Context observations for task inference. |
| context_actions | int32 | $(B, K)$ | Context actions. |
| context_next_obs | float32 | $(B, K, H, W, C)$ | Context next-state observations. |
| context_rewards | float32 | $(B, K)$ | Context rewards. |
| context_teammate_actions | int32 | $(B, K)$ | Context teammate actions (optional, for +TA). |

# E. Implementation Details of ICRL Baselines

This section provides comprehensive implementation details for the two ICRL baseline methods evaluated in our benchmark: Algorithm Distillation (AD) (Laskin et al., 2023) and Decision-Pretrained Transformer (DPT) (Lee et al., 2023). Our implementations are built entirely in JAX (Bradbury et al., 2018) and Flax (Heek et al., 2024), enabling seamless integration with the JAX-native OvercookedV2 environment (Gessler et al., 2025) and efficient GPU-accelerated training.

## E.1. Shared Architectural Components

Both AD and DPT share a common observation encoder designed for the grid-based partial observations in OvercookedV2.

**CNN Observation Encoder.** Following the architecture from JaxMARL (Rutherford et al., 2024), we employ a convolutional neural network to encode the spatial observations of shape $(H, W, C)$ where $H = W = 5$ (field of view) and $C$ varies by layout. The encoder consists of two stages:

1. **Pointwise Feature Extraction**: Three $1 \times 1$ convolutional layers with 128, 128, and 8 output channels respectively, each followed by ReLU activation. These layers perform channel-wise feature transformation without spatial mixing.

2. **Spatial Feature Extraction**: Three $3 \times 3$ convolutional layers with 16, 32, and 32 output channels respectively, each followed by ReLU activation. These layers capture local spatial patterns and object relationships.

The resulting feature map is flattened and projected through a dense layer to produce an embedding of dimension $d_{\text{emb}} = 64$. All convolutional and dense layers use orthogonal initialization (Saxe et al., 2014) with gain $\sqrt{2}$ and zero bias initialization.

**Transformer Architecture.** Both models employ a decoder-only Transformer (Vaswani et al., 2017) with pre-layer normalization (Xiong et al., 2020). Each Transformer block consists of:

- Multi-head self-attention with $h = 4$ heads and causal masking
- Feed-forward network with hidden dimension $4 \times d_{\text{hidden}}$ and GELU activation (Hendrycks, 2016)
- Residual connections and dropout (rate $0.1$) after each sub-layer

The default configuration uses $L = 4$ Transformer layers with hidden dimension $d_{\text{hidden}} = 256$. A final layer normalization is applied before the action prediction head.

## E.2. Algorithm Distillation (AD)

AD (Laskin et al., 2023) trains a sequence model to autoregressively predict actions from learning histories, enabling the model to internalize the learning dynamics of an RL algorithm and reproduce them at test time.

**Token Format.** At each timestep $t$, AD constructs a token by concatenating:

$$\mathbf{x}_t = [\text{Emb}(a_{t-1}^i); r_{t-1}; \text{Enc}(o_t^i)] \tag{11}$$

where $\text{Emb}(\cdot)$ denotes action embedding (learned embedding table with $d_{\text{emb}}$ dimensions), $r_{t-1}$ is the scalar reward from the previous timestep, and $\text{Enc}(\cdot)$ is the CNN observation encoder. The concatenated vector has dimension $2d_{\text{emb}} + 1$ and is projected to $d_{\text{hidden}}$ via a linear layer. For the first timestep, $a_{t-1}^i$ and $r_{t-1}$ are set to zero.

**Teammate Action Conditioning (+TA).** For the teammate-action-conditioned variant, we extend the token format to include the teammate's previous action:

$$\mathbf{x}_t^{+\text{TA}} = [\text{Emb}(a_{t-1}^i); \text{Emb}(a_{t-1}^{-i}); r_{t-1}; \text{Enc}(o_t^i)] \tag{12}$$

where $a_{t-1}^{-i}$ is the teammate's action at the previous timestep. This variant uses a separate learned embedding table for teammate actions with the same dimensionality.

**Training Objective.** The model is trained with cross-entropy loss on action prediction:

$$\mathcal{L}_{\text{AD}} = -\frac{1}{|\mathcal{M}|} \sum_{t \in \mathcal{M}} \log \pi_\theta(a_t^i | \mathbf{x}_{1:t}) \tag{13}$$

where $\mathcal{M}$ denotes the set of valid (non-padded) positions in the sequence, and $\pi_\theta$ is the action distribution output by the model. Causal attention masking ensures that predictions at position $t$ only attend to positions $\leq t$.

**Online Inference Buffer.**  During evaluation, AD maintains a rolling context buffer that is updated at *every timestep*. We implement this as a circular buffer with $O(1)$ insertion complexity, avoiding the $O(n)$ cost of array reallocation. The buffer stores tuples $(o_t^i, a_{t-1}^i, r_{t-1})$ and optionally $a_{t-1}^{-i}$ for teammate action conditioning. When the buffer reaches maximum capacity $K$, the oldest entries are evicted in FIFO order.

### E.3. Decision-Pretrained Transformer (DPT)

DPT (Lee et al., 2023) frames in-context learning as posterior sampling over tasks, training a Transformer to predict expert actions given a context of transitions that characterize the current task dynamics.

**Sequence Format.**  DPT uses a query-context architecture where the input sequence is:

$$[\mathbf{q}; \mathbf{c}_1; \mathbf{c}_2; \ldots; \mathbf{c}_K] \tag{14}$$

The query $\mathbf{q}$ contains only the current observation (with zeros for action, next observation, and reward), while each context element $\mathbf{c}_k$ represents a complete transition:

$$\mathbf{c}_k = [\text{Enc}(o_k^i); \text{OneHot}(a_k^i); \text{Enc}(o_{k+1}^i); r_k] \tag{15}$$

where $\text{OneHot}(\cdot)$ produces a one-hot vector of dimension equal to the action space size. The concatenated representation has dimension $2d_{\text{emb}} + |\mathcal{A}| + 1$ and is projected to $d_{\text{hidden}}$.

**Teammate Action Conditioning (+TA).**  The teammate-action-conditioned variant extends each context element:

$$\mathbf{c}_k^{\text{+TA}} = [\text{Enc}(o_k^i); \text{OneHot}(a_k^i); \text{OneHot}(a_k^{-i}); \text{Enc}(o_{k+1}^i); r_k] \tag{16}$$

This increases the transition embedding dimension by $|\mathcal{A}|$ additional dimensions.

**Training Objective.**  DPT requires expert action labels, which we obtain by relabeling learning histories with actions from the final converged policy. Following the original DPT formulation with prior, we compute loss on all sequence positions:

$$\mathcal{L}_{\text{DPT}} = -\frac{1}{K+1} \sum_{j=0}^{K} \log \pi_\theta(a^{i,*} | \mathbf{x}_{0:j}) \tag{17}$$

where $a^{i,*}$ is the expert action for the query observation. This "with prior" training encourages the model to predict the expert action even with partial context.

**Episode-Level Context Buffer.**  Unlike AD's step-level updates, DPT updates its context buffer only after each episode completes. The buffer stores complete transitions $(o^i, a^i, o^{i'}, r)$ from finished episodes. We implement this with a two-tier structure: a current-episode buffer that accumulates transitions within an episode, and a context buffer of completed episodes. Upon episode termination, transitions are moved from the current buffer to the context buffer using circular indexing for $O(1)$ amortized insertion.

### E.4. Training Infrastructure and Optimizations

We implement several optimizations to enable efficient training on large-scale learning history datasets.

**Learning Rate Schedule.**  Both models use a combined warmup and cosine decay schedule:

$$\eta(t) = \begin{cases} \eta_{\max} \cdot \frac{t}{T_{\text{warmup}}} & \text{if } t < T_{\text{warmup}} \\ \eta_{\max} \cdot \frac{1+\cos(\pi \cdot \frac{t - T_{\text{warmup}}}{T - T_{\text{warmup}}})}{2} & \text{otherwise} \end{cases} \tag{18}$$

where $\eta_{\max} = 10^{-3}$ is the peak learning rate, $T_{\text{warmup}}$ is 5% of total steps, and $T$ is the total training steps.

**Optimizer.** We use AdamW (Loshchilov & Hutter, 2019) with gradient clipping (max norm 1.0) implemented via Optax (DeepMind et al., 2020). Weight decay is set to 0.0 by default but can be configured.

**Threaded Data Prefetching.** To overlap I/O with GPU computation, we implement a background prefetching mechanism. A dedicated worker thread continuously samples batches from the HDF5 dataset and places them in a bounded queue (default size 5). The training loop retrieves batches from this queue, ensuring the GPU is never starved for data. This optimization provides significant speedup, particularly for large sequence lengths where batch construction involves substantial random access to the HDF5 file.

**HDF5 Caching.** We configure a large chunk cache (512 MB by default) for the HDF5 reader, reducing disk I/O for repeated access patterns during training. The cache size can be adjusted based on available system memory.

**Asynchronous Device Transfer.** We leverage JAX's asynchronous dispatch to overlap CPU-to-GPU data transfer with computation. Batch data is placed on the GPU device immediately after construction, allowing the transfer to proceed concurrently with the previous training step's gradient computation.

**JIT Compilation.** Training and evaluation step functions are JIT-compiled with static argument specification for configuration parameters (number of actions, label smoothing, etc.), enabling XLA fusion and optimization while avoiding recompilation when hyperparameters remain constant.

### E.5. Hyperparameter Summary

Table 22 summarizes the default hyperparameters used for both AD and DPT in our experiments.

*Table 22.* **Default Hyperparameters for AD and DPT.** Both methods share most architectural and training hyperparameters, differing primarily in their sequence format and loss computation.

| Hyperparameter | AD | DPT |
|---|---|---|
| *Architecture* | | |
| Observation embedding dim | 64 | 64 |
| Transformer hidden dim | 256 | 256 |
| Number of layers | 4 | 4 |
| Number of attention heads | 4 | 4 |
| Context length (transitions) | 500 | 500 |
| *Regularization* | | |
| Attention dropout | 0.1 | 0.1 |
| Residual dropout | 0.1 | 0.1 |
| Embedding dropout | 0.1 | 0.1 |
| Label smoothing | 0.0 | 0.0 |
| *Training* | | |
| Batch size | 1024 | 1024 |
| Learning rate | $10^{-3}$ | $10^{-3}$ |
| Warmup steps | 5% of total | 5% of total |
| Weight decay | 0.0 | 0.0 |
| Gradient clip norm | 1.0 | 1.0 |
| Training steps | 20,000 | 20,000 |
| *Method-Specific* | | |
| Buffer update granularity | Step-level | Episode-level |
| Requires expert labels | No | Yes |
| Loss computation | All positions | With prior |

### E.6. Code Availability

Our implementation is released as part of the ICRL4AHT benchmark. The codebase includes:

- Complete AD and DPT model implementations in Flax
- Training scripts with configurable hyperparameters
- Online evaluation buffers with efficient circular indexing
- Integration with the learning history dataset format
- Checkpoint saving and resumption support

The code will be publicly available upon publication to facilitate reproducibility and extension by the research community.

# F. Additional Experiments and Analyses

### F.1. Trajectory Filtering for Learning History Curation

A critical innovation in our data pipeline is trajectory filtering, which significantly improves the quality of learning histories for AD-style ICRL training. During data collection, we execute PPO training across multiple parallel environment streams simultaneously, where each stream represents an independent learning trajectory against a fixed teammate. Raw learning curves exhibit substantial stochasticity due to exploration noise, environment randomness, and the inherent variance of policy gradient methods. These noisy trajectories pose challenges for sequence models attempting to learn the underlying improvement dynamics.

To address this, we implement a quality-based filtering procedure that selects the best environment streams based on a composite quality score. Specifically, for each environment stream with episode returns $\{R_1, R_2, \ldots, R_N\}$, we compute window-averaged returns at the beginning and end of training. Let $\bar{R}_{\text{start}}$ denote the mean return over the first $w$ episodes and $\bar{R}_{\text{end}}$ denote the mean over the final $w$ episodes, where $w = \max(5, \lfloor 0.1N \rfloor)$. Quality score for each stream is defined as:

$$\text{score} = \bar{R}_{\text{end}} + (\bar{R}_{\text{end}} - \bar{R}_{\text{start}}) \tag{19}$$

This score jointly rewards high final performance and substantial improvement over the training horizon. We then rank all environment streams by their quality scores and retain only the top-$k$ streams per task, where $k$ is a configurable parameter. The default setting in our experiments is $k = 128$.

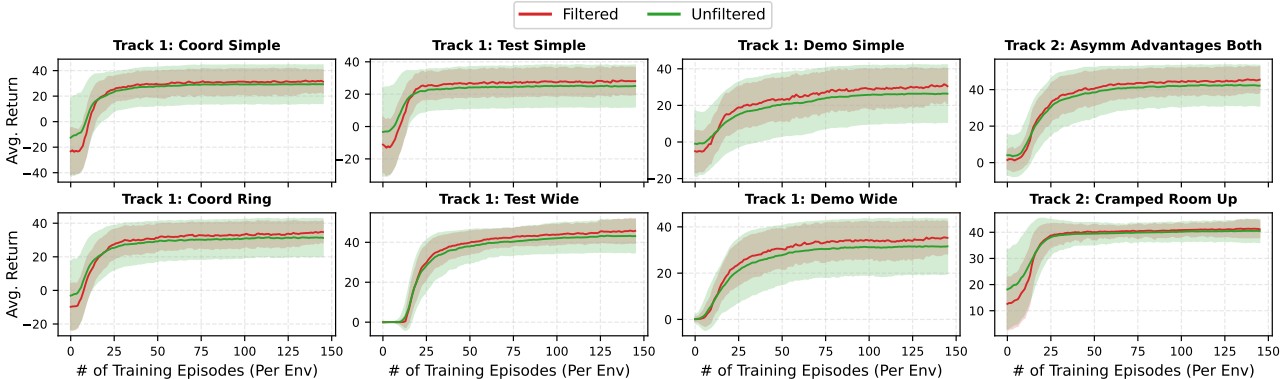

*Figure 8.* **Source Algorithm Training Curves.** Evaluation returns of PPO-based best-response policies trained against fixed teammates during learning-history dataset construction. We compare raw trajectories (before filtering) with filtered trajectories selected based on quality scores that reward high final performance and large improvement. The filtered curves exhibit steeper improvement gradients and higher final performance, validating our data curation strategy for eliciting in-context learning.

As illustrated in Fig. 8, this filtering procedure yields learning histories with substantially steeper improvement gradients and reduced variance. The filtered curves more clearly demonstrate the progressive skill acquisition pattern that AD is designed to model and reproduce during test-time adaptation. Empirically, this curation strategy reduces the dataset size from 1.19B to approximately 150M transitions while dramatically improving the signal-to-noise ratio, ensuring that the ICRL Transformer learns to model genuine improvement dynamics rather than fitting to stochastic fluctuations.

## F.2. Teammate Policy Diversity Analysis

To validate that our teammate suite induces meaningful distribution shifts between training and testing, we quantify behavioral diversity using pairwise Hamming distance. This metric measures the proportion of states in which two policies select different actions, providing a direct assessment of behavioral dissimilarity.

**Hamming Distance Computation.** For two policies $\pi_A$ and $\pi_B$, we define the Hamming distance as:

$$d_H(\pi_A, \pi_B) = 1 - \frac{1}{N} \sum_{s \in \mathcal{S}} \mathbb{1}\{\pi_A(s) = \pi_B(s)\} \tag{20}$$

where $\mathcal{S}$ is a set of $N$ sampled states and $\mathbb{1}\{\cdot\}$ is the indicator function. A Hamming distance of 0 indicates identical action selection across all states, while 1 indicates completely divergent behavior.

To obtain a representative state distribution, we execute rollouts across all layouts using a fixed random policy for agent 0, while cycling through teammate policies for agent 1. At each timestep, we query all policies for their action given the current observation. This procedure ensures that the state distribution reflects realistic interaction dynamics rather than artificial corner cases.

**Inter-Family Diversity.** We treat all RL-trained policies as a single family $\mathcal{F}_{\text{RL}}$ (comprising 80 distinct policies from various algorithms and checkpoints) and compute pairwise Hamming distances among five families: $\mathcal{F}_{\text{RL}}$ and the four heuristic families H1–H4 (each with 5 parameter configurations). For each pair of families $\mathcal{F}_A$ and $\mathcal{F}_B$ with policy sets $\Pi_A$ and $\Pi_B$, we compute the average Hamming distance:

$$\bar{d}_H(\mathcal{F}_A, \mathcal{F}_B) = \frac{1}{|\Pi_A||\Pi_B|} \sum_{\pi_a \in \Pi_A} \sum_{\pi_b \in \Pi_B} d_H(\pi_a, \pi_b) \tag{21}$$

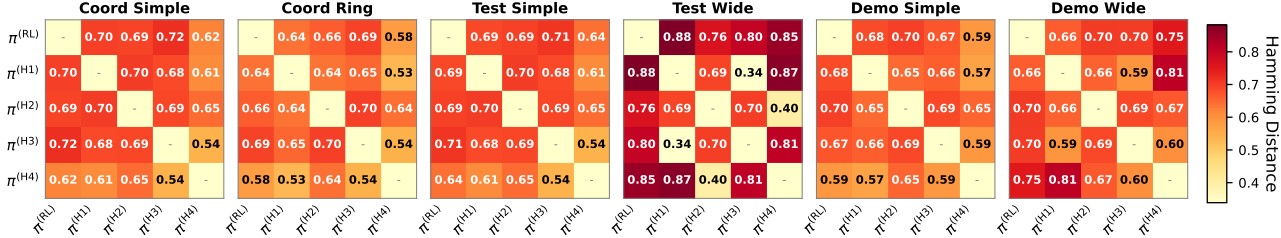

*Figure 9.* **Teammate Policy Diversity.** Pairwise Hamming distance heatmaps among five policy families: RL-trained policies ($|\mathcal{F}_{\text{RL}}| = 80$) and four heuristic families H1–H4 ($|\mathcal{F}_{\text{H}i}| = 5$ each). High inter-family distances, particularly between RL and heuristic families, confirm that our train-test split induces substantial behavioral distribution shift, ensuring that generalization to held-out heuristic families requires genuine in-context inference.

As shown in Fig. 9, the RL-trained policies exhibit consistently high Hamming distances from all four heuristic families across benchmark layouts. The inter-family distances between $\mathcal{F}_{\text{RL}}$ and heuristic families typically exceed 0.5, indicating that RL and heuristic policies select different actions in the majority of encountered states. This behavioral divergence validates our experimental design: since ICRL agents are trained exclusively on RL-generated learning histories, adaptation to held-out heuristic families in $\Pi_{\text{test}}$ cannot be achieved through interpolation within the training distribution, but instead requires genuine in-context inference of the partner's behavioral patterns.

### F.3. Teammate Action Conditioning: Online Adaptation Curves

In the main text (Table 5), we presented aggregate performance metrics for the teammate action conditioning ablation (+TA variants). Here we provide the corresponding online adaptation curves to examine whether providing explicit access to teammate actions enables the emergence of in-context learning dynamics.

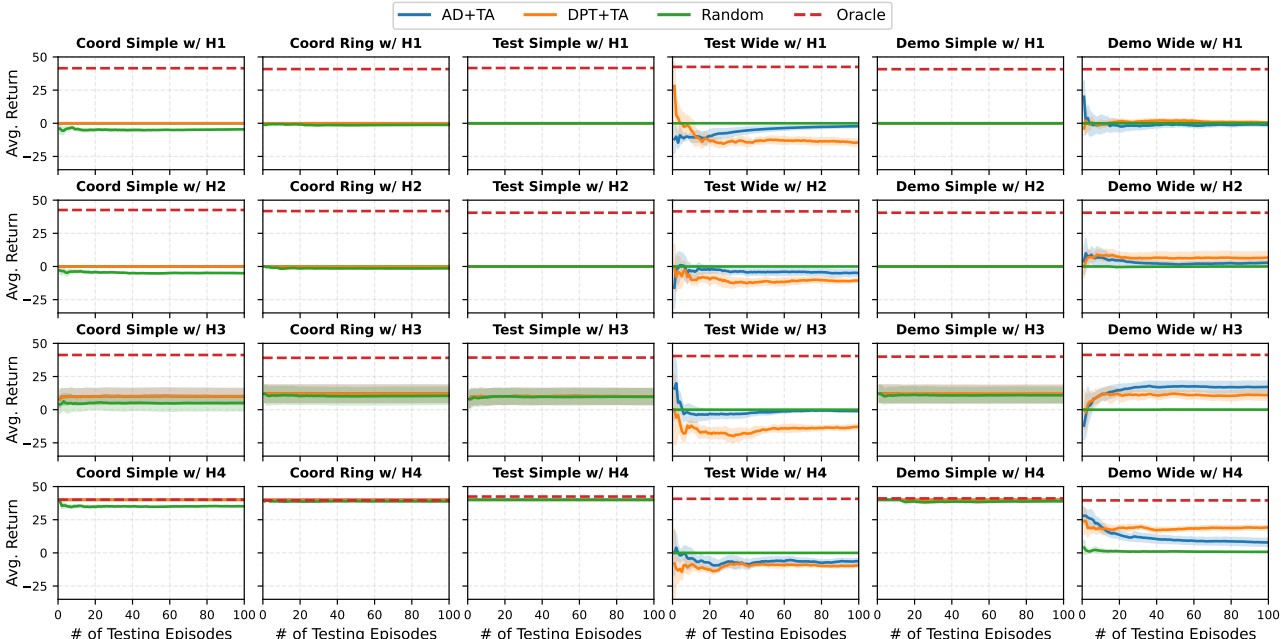

*Figure 10.* **Teammate Action Conditioning: Online Adaptation Curves** ($\mathcal{L}_{\text{train}} \times \Pi_{\text{test}}$). Episode-wise return trajectories of AD+TA, DPT+TA, and random baseline across six training layouts and four heuristic teammate families. Despite conditioning on ground-truth teammate actions, the +TA variants exhibit adaptation profiles qualitatively similar to their unconditional counterparts (Fig. 3): returns remain flat across the interaction horizon with no discernible upward trend. This visualization confirms that the absence of in-context improvement is not primarily attributable to partial observability of partner behavior.

Fig. 10 presents the episode-wise return trajectories for AD+TA and DPT+TA across all Track 1 layouts and heuristic families. Several observations emerge from this analysis:

**Persistent Absence of Adaptation.** Analogous to the unconditional variants shown in Fig. 3, the +TA models exhibit flat adaptation profiles throughout the evaluation horizon. Returns neither increase systematically as more episodes accumulate nor show evidence of the characteristic "elbow" pattern observed in successful single-agent ICRL, where performance improves rapidly after an initial exploration phase. This finding is particularly striking given that teammate actions provide direct, unambiguous evidence of partner behavior—information that should, in principle, substantially accelerate policy identification.

**Layout-Dependent Effects.** The impact of teammate action conditioning varies across layouts. On **test wide**, where unconditional methods exhibit severe negative returns, the +TA variants show modest improvement but remain substantially below the random baseline. Conversely, on layouts where unconditional methods already achieve reasonable performance (*e.g.*, **coord simple** with H4), teammate action conditioning provides negligible additional benefit. This pattern suggests that access to teammate actions may mitigate catastrophic interference but does not enable genuine adaptation.

**Implications for Architecture Design.** The failure of teammate action conditioning to elicit in-context learning suggests that the bottleneck lies not in information availability but in the model's capacity to extract and utilize partner-relevant features from the context. Standard Transformer architectures, trained with action prediction objectives, may lack the inductive biases necessary to perform the implicit partner modeling required for ad-hoc coordination. This motivates future investigation of architectures with explicit partner inference modules, as discussed in Sec. G.

## F.4. Supplementary Diagnostic Experiments

To further deepen our analysis and provide a more comprehensive empirical picture, we conduct a series of additional diagnostic experiments that systematically probe the robustness of our main findings under varied conditions. All experiments below use the same evaluation protocol and aggregation scheme described in Sec. 5 unless otherwise noted.

### F.4.1. HELD-OUT RL TEAMMATES

To establish an in-distribution generalization baseline, we evaluate AD and DPT on held-out RL-trained teammates rather than the heuristic test families. As shown in Fig. 11, performance on familiar layouts is moderately better under this milder shift, compared to the heuristic benchmark. However, strong within-context improvement remains absent. This indicates that current baselines struggle even under relatively mild teammate shift, and the cross-family heuristic evaluation in the main text represents a more severe but qualitatively consistent stress test.

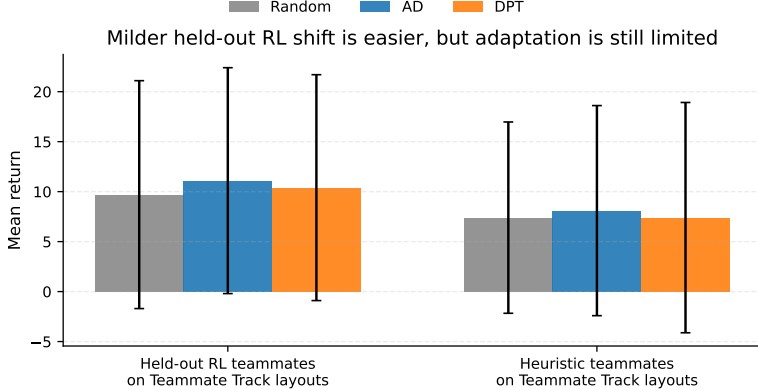

*Figure 11.* **Held-Out RL Teammate Evaluation.** AD and DPT evaluated on held-out RL-trained teammates on familiar layouts. Performance is moderately better than under cross-family heuristic shift, but within-context adaptation remains weak.

### F.4.2. EXTENDED CONTEXT LENGTHS

Our main evaluation uses context windows up to $K{=}2{,}000$ transitions. To investigate whether longer context enables adaptation, we extend the evaluation to $K \in \{500, 1{,}000, 2{,}000, 5{,}000, 10{,}000\}$. As shown in Fig. 12, longer context provides some benefit on moderately difficult settings but does not qualitatively change the outcome on the hardest configurations. For example, on the challenging `test wide` layout, AD improves marginally from $-7.9$ at $K{=}500$ to $-7.6$ at $K{=}1{,}000$ and $-8.8$ at $K{=}2{,}000$, while DPT goes from $-15.9$ to $-13.8$ and $-8.7$, with only modest improvement at $K{=}5{,}000/10{,}000$. The Transformer attention mechanism does not appear to leverage extended histories for effective partner inference in the hardest settings.

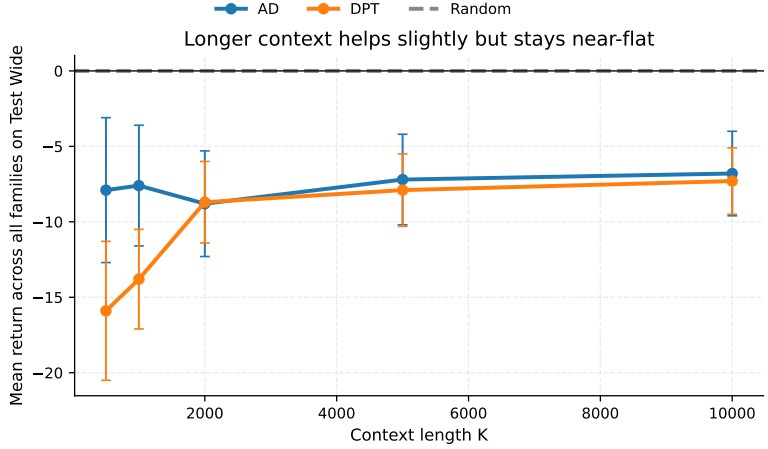

*Figure 12.* **Extended Context Length Evaluation.** Performance across $K \in \{500, 1{,}000, 2{,}000, 5{,}000, 10{,}000\}$. Longer context helps somewhat on moderately difficult settings but does not qualitatively reverse the negative result on the hardest configurations.

### F.4.3. AUXILIARY LOSS VARIANTS

We investigate whether auxiliary training objectives can improve adaptation. Three variants are evaluated: (1) **+TA**: conditioning on ground-truth teammate actions (as in the main text ablation); (2) **+next_obs**: auxiliary next-observation prediction loss; and (3) **mixed**: combining both teammate action conditioning and next-observation prediction. As shown in Fig. 13, the auxiliary variants produce modest stabilization on familiar teammate settings but remain weak on Layout Generalization. The +TA variant improves slightly on the Teammate Generalization track (AD: $8.10 \rightarrow 9.01$ mean return) but degrades on the Layout Generalization track (AD: $2.57 \rightarrow -0.45$). These results suggest that the adaptation bottleneck is deeper than a single missing auxiliary objective.

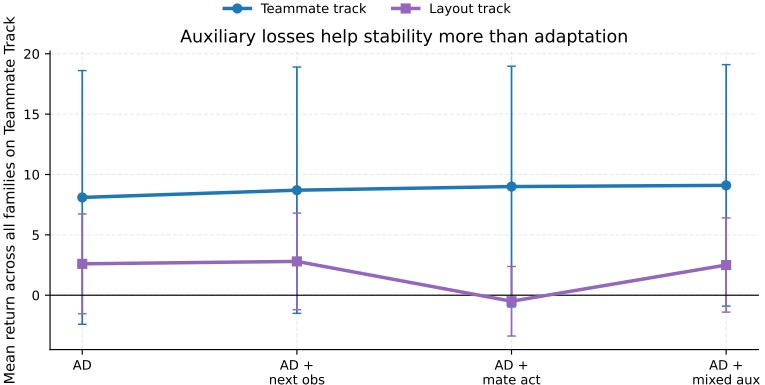

*Figure 13.* **Auxiliary Loss Variants.** Performance comparison of +TA, +next_obs, and mixed auxiliary objectives. Auxiliary losses provide modest stabilization on familiar settings but do not resolve the layout generalization failure.

### F.4.4. WITHIN-ROLLOUT ADAPTATION GAIN

To directly quantify the degree of online adaptation, we compute the *adaptation gain*: the difference between the mean return over the last 20 episodes and the mean return over the first 20 episodes within the same evaluation rollout, aggregated by track and algorithm. If a method is successfully adapting to its partner over the course of interaction, this quantity should be clearly positive. As shown in Fig. 14, the adaptation gains remain small on average across both tracks and all baselines, confirming that the issue is not merely a short warm-up period but reflects weak within-rollout adaptation overall. This metric provides complementary evidence to the episode-wise curves in the main text, isolating the temporal adaptation signal from absolute performance level.

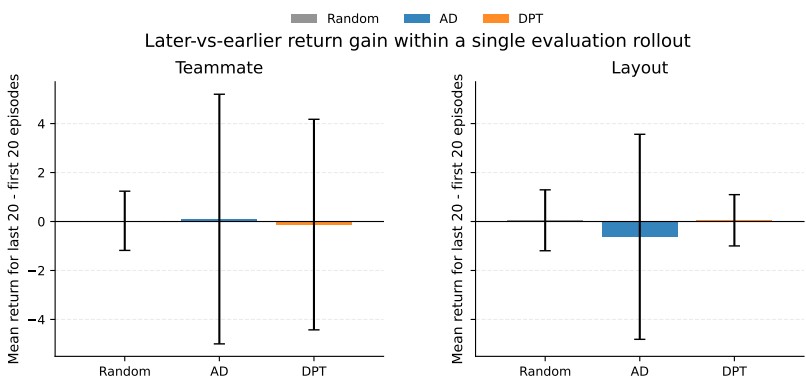

*Figure 14.* **Within-Rollout Adaptation Gain** ($\Delta = \bar{R}_{\text{last 20}} - \bar{R}_{\text{first 20}}$). Adaptation gains are small on average across all baselines and both tracks, confirming weak within-rollout online adaptation.

# G. Limitations and Future Work

This section provides an extended discussion of the limitations of ICRL4AHT and outlines promising directions for future research.

## G.1. Benchmark Scope

While ICRL4AHT provides a rigorous testbed for evaluating in-context adaptation in cooperative settings, several scope limitations warrant acknowledgment.

**Domain Specificity.** Our benchmark is instantiated exclusively on OvercookedV2, a domain that, despite its strategic richness and partial observability, represents only one point in the space of cooperative multi-agent problems. OvercookedV2 features discrete action spaces, grid-world dynamics, and relatively short episode horizons. The degree to which our findings generalize to other coordination domains—such as Hanabi's information asymmetry and convention formation, Melting Pot's mixed-motive scenarios with larger populations, or continuous control tasks like cooperative manipulation—remains an open empirical question. Future benchmark extensions should incorporate diverse coordination substrates to establish whether the observed ICRL failures are domain-specific or reflect fundamental architectural limitations.

**Teammate Diversity.** Our teammate suite, though designed for diversity, is finite. The four heuristic families (H1–H4) span a spectrum of cooperability levels but may not exhaustively cover the space of possible partner behaviors encountered in real-world deployment. In particular, adversarial or deceptive partners, partners with non-stationary policies, and partners exhibiting complex hierarchical behaviors are not represented. Additionally, the RL-trained policies, while generated via multiple algorithms (FCP, BRDiv, LBRDiv, CoMeDi), share common training substrates and may exhibit correlated failure modes that do not stress-test generalization as thoroughly as truly independent policy sources would.

**Team Structure.** We restrict evaluation to two-player settings with fixed, non-adaptive partners. This design choice isolates the challenge of unilateral adaptation but leaves open several questions: (1) Can ICRL scale to larger teams where the combinatorial complexity of partner inference grows exponentially? (2) How does performance degrade when partners themselves exhibit learning dynamics, creating non-stationary effective MDPs? (3) Can ICRL handle asymmetric information structures where different agents have access to different observation modalities? Extending the benchmark to $n$-player settings with co-adapting partners would provide a more comprehensive assessment of ICRL capabilities.

## G.2. Methodological Constraints

Our experimental design imposes constraints that, while necessary for controlled evaluation, future work should systematically relax.

**Distribution Separation.** The strict separation of RL-trained policies for training and heuristic policies for testing ensures that test-time performance reflects genuine generalization rather than memorization. However, this design may not reflect realistic deployment scenarios where training and test distributions overlap partially or where agents encounter partners drawn from a continuum of behavioral types. Investigating ICRL performance under varying degrees of distribution shift—from interpolation within the training distribution to extreme extrapolation—would provide finer-grained insights into the nature of generalization failures.

**Context Length Bounds.** Our main evaluation focuses on context windows up to $K = 2000$ transitions, corresponding to approximately 20 episodes. Extended evaluation up to $K = 10{,}000$ (Sec. F.4.2) shows modest improvement on selected settings but does not qualitatively change the outcome on the hardest configurations. While this exceeds typical single-agent ICRL evaluations, it may be insufficient for the more challenging inference problem posed by multi-agent coordination. Theoretical analysis suggests that identifying a partner's policy from behavioral observations may require substantially more data than identifying environment dynamics in single-agent settings, particularly when the partner's policy space is large or when behaviors are only weakly distinguishable.

**Observation Assumptions.** The current pipeline assumes access to ground-truth teammate actions during data collection for the +TA ablation variants. This assumption may not hold in settings with communication constraints, asynchronous execution, or partial observability of partner actions. Furthermore, our observation model provides a fixed $5 \times 5$ field

of view, which may be either too restrictive (preventing observation of relevant partner behaviors) or too permissive (providing information that would be unavailable in more realistic settings). Systematic variation of observation fidelity would illuminate the relationship between information availability and adaptation capability.

**Evaluation Metrics.**   We primarily report episodic returns averaged across teammates and layouts. While this provides a summary measure of coordination capability, it may obscure important aspects of behavior such as: (1) the variance of performance across episodes, which indicates consistency; (2) the speed of adaptation within a context window; (3) the degree to which agents exhibit interpretable coordination strategies versus achieving returns through unintended mechanisms. Richer evaluation protocols incorporating behavioral analysis, strategy identification, and human interpretability assessments would provide deeper insights.

### G.3. Future Directions

Our findings reveal that the representative history-conditioned ICRL baselines studied here (AD, DPT, and their variants) fail to perform the strategic inference required for ad-hoc coordination within the OvercookedV2 AHT protocol. The supplementary experiments in Sec. F.4—including hold-out RL teammates, extended context, and auxiliary losses—are diagnostic in nature rather than an exhaustive leaderboard; they serve to confirm the robustness of the main finding under reasonable variations. We identify three primary research directions to address this gap.

**Explicit Partner Modeling.**   The core limitation exposed by our experiments is the inability of standard Transformers to infer partner policies from interaction histories. We hypothesize that architectural innovations incorporating explicit partner representations are necessary (Jing et al., 2024b). Promising approaches include: (1) *latent teammate inference*, where variational modules infer a distribution over partner types that conditions action selection; (2) *theory-of-mind attention*, where dedicated attention heads model partner state separately from environment dynamics; and (3) *auxiliary prediction objectives* that explicitly supervise partner behavior prediction, forcing the model to develop causal understanding of how partner actions influence outcomes.

**Structured Training for Coordination.**   Current ICRL methods train on action prediction loss alone, which may be insufficient for learning the exploration and identification skills required in AHT. Future work should investigate training regimes that explicitly incentivize: (1) *information-seeking behavior* early in interaction to accelerate partner identification; (2) *curriculum learning* that progressively exposes agents to more diverse and challenging partners; and (3) *multi-task coordination* across diverse environments to encourage learning of transferable partner modeling capabilities rather than layout-specific heuristics.

**Human-Centric Evaluation.**   While ICRL4AHT provides systematic evaluation against algorithmic partners, the ultimate goal of AHT is coordination with humans. Extending our benchmark to incorporate human interaction data—through large-scale human-human gameplay collection and human proxy models—would enable evaluation of whether ICRL methods can generalize to the nuanced, context-dependent behaviors exhibited by human partners. Such extension would also illuminate whether effective human-AI coordination requires interpretable agent strategies that humans can predict and complement.

**Extension to LLM-Based Agents.**   Another promising direction is to extend ICRL4AHT to evaluate large language model (LLM)-based agents (Cao et al., 2026; Wang et al., 2026). Compared with sequence models trained solely on domain-specific learning histories, pretrained LLMs may provide broader behavioral priors, explicit reasoning capabilities, and flexible mechanisms for maintaining and interpreting long interaction histories (Jing et al., 2026). By translating environment observations and actions into textual or multimodal interfaces, future work could investigate whether LLM agents can infer latent teammate strategies, formulate and revise coordination plans, and adapt their behavior through prompting, memory, or lightweight fine-tuning. The benchmark could also be extended to settings with natural-language communication, enabling systematic study of when communication improves coordination and when agents must instead infer partner intent from behavior alone. Such evaluations would help determine whether the limitations identified in this work arise primarily from current ICRL training paradigms or persist even for general-purpose foundation models with substantially richer prior knowledge and reasoning capacity.

