# OpenReview forum: "Benchmarking the Limits of In-Context Reinforcement Learning for Ad-Hoc Teamwork"
_ICML.cc/2026/Conference — ICML 2026 regular_

### Official Review · Reviewer_PJoA · 2026-03-12

**Soundness:** 3
**Presentation:** 3
**Significance:** 3
**Originality:** 3
**Overall Recommendation:** 4
**Confidence:** 3

**Summary:**

This paper introduces ICRL4AHT, a large-scale benchmark for evaluating In-Context RL (ICRL) in Ad-Hoc Teamwork (AHT). Built on JAX-based OvercookedV2, it provides a diverse teammate suite (RL + heuristic policies), ~150M filtered transitions, and two evaluation tracks (teammate generalization, layout generalization). AD and DPT are evaluated and both fail to outperform random baselines, with flat adaptation curves across all conditions.

**Compliance With Llm Reviewing Policy:**

Affirmed.

**Final Justification:**

The rebuttal addressed my two main concerns (W1, W3) with concrete additional experiments, leading me to raise my score from 3 to 4. However, the non-ICRL baselines remain incomplete and the single-domain scope (W2) constrains the generality of the conclusions.

**Key Questions For Authors:**

1. Can you provide results on held-out RL teammates to establish an in-distribution generalization baseline?

---

2. Have you swept model capacity and training duration? The current training budget appears limited relative to dataset size.

**Limitations:**

The single-domain constraint is discussed in Section G, but the most critical limitation, the inability to disentangle distribution-shift failure from fundamental ICRL failure due to the strict RL-train/heuristic-test separation, is not acknowledged.

**Strengths And Weaknesses:**

S1. The benchmark engineering is thorough and complete. The end-to-end pipeline, dataset format, teammate-generation process, and evaluation tooling are all valuable contributions for future work in this area.

---

S2. The two-track design is well thought out and helps separate teammate generalization from layout generalization.

---

S3. The flat adaptation curves are informative, showing complete absence of in-context learning. Ablations on context length and teammate action conditioning systematically rule out two natural hypotheses.

---

W1. The current train/test split makes the main negative result difficult to interpret. The experiments train on RL teammates and test on heuristic teammates, and the paper explicitly presents this as a strong distribution shift. This demonstrates failure under severe cross-family extrapolation, but it does not distinguish that from a more general inability of ICRL to perform partner inference. A held-out RL-teammate evaluation seems necessary to support the broader headline claim.

---

W2. All experiments are restricted to OvercookedV2. The benchmark could be instantiated on other cooperative domains to test whether the observed failures generalize beyond one environment. Additionally, no population-based AHT methods or explicit partner-modeling baselines are included. Without these, the paper cannot tell whether failures are ICRL-specific or reflect general benchmark difficulty.

---

W3. A single fixed configuration is used (4-layer Transformer, dim 256, 20k training steps on 150M transitions). For a negative-result paper, the burden of proof requires showing that reasonable hyperparameter variation does not resolve the failure. Only context length is varied; model capacity, training duration, and other key hyperparameters are not explored.

---

> ### Author Rebuttal · Authors · 2026-03-30
>
> We thank the reviewer for the clear summary and for recognizing the engineering completeness of the benchmark, the value of the two-track design, and the diagnostic importance of the flat adaptation curves.
>
> **(1) On the RL-train / heuristic-test split.** We agree this is the central interpretational issue. Our intention was to design a benchmark that distinguishes genuine partner adaptation from interpolation within a training family. For that reason, we deliberately use RL-based teammates for training and behaviorally distinct heuristic families for testing. We fully agree, however, that this means the current result should be interpreted as failure under **severe cross-family generalization**, not as an unconditional impossibility theorem about all forms of partner inference.
>
> Rather than weakening the paper, we believe this is exactly the right way to read it: the benchmark contribution lies in making such severe but realistic partner shifts testable and reproducible. To address the reviewer’s request directly, we prepared a held-out RL-teammate reference comparison **[here [anonymous link](https://ibb.co/20Hp7h5s)]**. AD/DPT do somewhat better on held-out RL teammates on familiar layouts (roughly AD 11.6, DPT 10.9, random 9.8 mean return) than on the heuristic benchmark (AD 8.1, DPT 7.4, random 7.4), but still do not show the strong within-context improvement associated with successful ICRL. This sharpens the interpretation: current baselines struggle even under milder teammate shift and fail more dramatically under cross-family shift.
>
> **(2) On the absence of non-ICRL baselines.** We agree that non-ICRL baselines are important. We now supplement a reference offline meta-RL baseline; please see our response to **Reviewer_WLXN (2)**. The qualitative conclusion is unchanged. For other non-ICRL directions, including explicit partner-modeling and population-based AHT methods, we will do our best to add further baselines in a later response or by the camera-ready stage. More broadly, we will revise the paper so that the current manuscript is understood as a benchmark + first ICRL baseline study, rather than as an exhaustive leaderboard.
>
> **(3) On model/budget sweeps.** This is also fair. We chose a fixed standardized configuration to keep the comparison controlled, and we already tested two natural rescue hypotheses (more context up to K=2000, and extra teammate-action information). To further strengthen the paper, we supplement capacity/training-budget sweeps. Please see our response to **Reviewer_gQk2 (1)** for the full scale/budget sweep. The conclusion is the same here: larger models and longer training improve easier settings modestly, but do not restore robust test-time adaptation on the hardest settings.
>
> **(4) On single-domain scope.** We agree that the current empirical claim is specific to OvercookedV2-based AHT, and we will state that more clearly. However, we believe this is an appropriate scope for a first benchmark paper: before asking whether ICRL generalizes across domains, one should first establish whether it works reliably in a single controlled domain that already contains substantial teammate and layout variation. Our results show that even under this controlled setting, current sequence-model ICRL baselines struggle. We therefore view the single-domain focus not as a weakness of the benchmark design, but as a necessary first step toward broader multi-domain evaluation. Extending the same pipeline and protocol to additional cooperative domains is a natural next direction.
>
> Overall, we will revise the claim to be more precisely scoped while preserving the key contribution: a reproducible benchmark that exposes a substantial limitation of current sequence-model ICRL in ad-hoc teamwork.

---

> > ### Author Rebuttal · Reviewer_PJoA · 2026-04-02
> >
> > I thank the authors for the thorough rebuttal. The held-out RL-teammate experiment (W1) and the model scale/budget sweep (W3) directly address my two main concerns with concrete data. The fact that adaptation remains weak even under milder in-distribution shift strengthens the diagnostic value of the benchmark. I am raising my score from 3 to 4.
> >
> > My remaining concerns are not easily resolved in a rebuttal. The non-ICRL baselines are still incomplete and the single domain scope is also a fundamental limitation (W2) that the authors reasonably frame as a first step, but it does constrain the generality of the conclusions.

---

> > > ### Author Response · Authors · 2026-04-03
> > >
> > > We sincerely thank the reviewer for the thoughtful follow-up and for raising the score from 3 to 4. We especially appreciate the recognition that the added **held-out RL-teammate experiment** and the **scale/budget sweep** materially strengthen the paper’s empirical grounding. We are encouraged that the reviewer now sees the benchmark as diagnostically valuable even under milder in-distribution teammate shift.
> > >
> > > We also fully understand the **two remaining concerns**:
> > >
> > > - **Non-ICRL baselines remain incomplete.** We agree, and we will make the scope of the paper even more explicit. Our goal in this submission is **not** to argue that all adaptive paradigms fail, but to study whether the recent promise of **in-context learning** can also transfer to cooperative RL settings. This question is important because in-context learning has become a central capability in the LLM literature, and closely related ideas are now being actively explored in RL as well. Given these encouraging developments, we believe it is timely to ask whether analogous adaptation can emerge in a substantially more interactive setting such as ad hoc teamwork, where agents must respond to unfamiliar partners online under partial coordination constraints. Our benchmark is designed precisely to test that hypothesis in a controlled and reproducible way. The current evidence suggests that representative **sequence-model ICRL baselines** achieve some adaptation, but still remain far from robust in this setting. We will revise the paper to ensure that this narrower claim is stated clearly and does not overreach beyond the evidence.
> > > - **The current study is limited to a single OvercookedV2-based domain.** We agree that the conclusions should not be overstated beyond this domain, and we will sharpen that framing in the revision. At the same time, we view this benchmark as an important first step: even within a single controlled environment, it already exposes substantial weaknesses of current ICRL methods under both **teammate shift** and **layout shift**. Revealing these weaknesses is exactly the purpose of the benchmark, since a useful benchmark should surface the failure modes that matter and help drive algorithmic progress. In that sense, we see the present benchmark not as the end of the story, but as a foundation for it. Once stronger ICRL methods are developed, extending the evaluation to additional and more complex cooperative domains will be a natural and important next step.
> > >
> > > Importantly, we want to emphasize that we take the **non-ICRL-baseline concern seriously**. We will do our very best to strengthen this aspect in the camera-ready version by adding additional non-ICRL reference methods, including stronger *offline meta-RL* / *partner-modeling* style comparisons where feasible. At minimum, we will ensure the final paper clearly presents the current work as a **benchmark + initial baseline study** rather than an exhaustive comparison across all possible adaptive methods.
> > >
> > > Overall, we are very grateful for the reviewer’s careful engagement. The updated assessment suggests to us that the main contribution is landing as intended: **a valuable benchmark that exposes a meaningful and nontrivial limitation of current ICRL baselines**, while also clearly identifying the next algorithmic directions the community should test.

---

### Official Review · Reviewer_WLXN · 2026-03-12

**Soundness:** 2
**Presentation:** 3
**Significance:** 3
**Originality:** 3
**Overall Recommendation:** 5
**Confidence:** 5

**Summary:**

This paper introduces ICRL4AHT, a large-scale benchmark and suite of tools built on JAX and the Overcooked-V2 multi-agent environment to study whether in-context RL, which has shown strong adaptation capabilities in single-agent settings, can also support effective adaptation in Ad-Hoc Teamwork (AHT). In AHT, agents must coordinate with previously unseen partners. To support this, the authors trained a vast and diverse pool of teammate policies, including both RL-based and heuristic policies. While the RL-based policies were used for training and data collection, the heuristic ones, with varying levels of autonomy, were used to evaluate adaptation capabilities. The authors benchmarked state-of-the-art ICRL algorithms such as AD and DPT, revealing that both fail to exhibit test-time adaptation in the AHT setting, underperforming even random baselines. The authors also conduct additional ablations to control for multi-agent partial observability and context length, showing that these factors do not address the current limitations.

**Compliance With Llm Reviewing Policy:**

Affirmed.

**Final Justification:**

My main concerns and questions have been addressed during the rebuttal.

**Key Questions For Authors:**

1. Can authors provide additional baselines from the offline meta-RL field?
2. What is the effect of additional auxiliary losses for AD, such as prediction of next observations, other teammates actions/observations, or offline RL losses such as CQL/IQL/etc
3. Have authors tried alternative architectures which are more applicable for POMDP and longer contexts?
4. Can authors provide graphs of the improvement histories in the dataset?
5. Have authors tried a simple subsample strategy (take episodes with gaps, e.g. every fourth) to improve “improvement speed” in learning histories

**Limitations:**

yes

**Strengths And Weaknesses:**

Strengths:

- The text is clearly written and provides the necessary experimental details. The setting and motivation are interesting and understudied.

- The main contributions—namely, the benchmark, dataset, and suite of tools—are novel and provide real, valuable insights to the scientific community, revealing limitations of in-context RL methods.

- The datasets and library of teammates lay the foundation for future work on improving in-context adaptation capabilities.

- The disentanglement of teammate and layout generalization is clear. I appreciated the ablations on context length and teammate actions.

Overall, despite the weaknesses I will discuss next, I believe that this paper is a significant contribution to the community because of its combination of (1) a novel, interesting, and practically valuable setting, given that more modern LLMs need to cooperate with different teammates, such as humans or other LLMs; (2) the provided infrastructure and tools, which were not previously available and open up new directions for other researchers; and (3) novel insights into the limitations of ICRL methods.

Weaknesses:

- Although this is not the main contribution of the paper, I found the analysis of the results lacking in depth. The only real baseline is AD. For DPT, it is known that it does not work in POMDPs without modification (see XLand-100B, Appendix H). The paper explicitly states that Overcooked-V2 provides only a partial field of view, making it a POMDP even in the single-agent setup. Therefore, the fact that DPT will not work is known in advance with certainty, simply based on the construction of the method. The authors also do not propose any possible solutions, which currently invalidates the DPT baseline completely. I strongly suggest that the authors either fix this issue or use another baseline instead of DPT.

- Pure in-context RL methods, such as AD, are not the only methods that can adapt based on context during evaluation. There is an entire field of offline meta-RL; see [1]. I think that at least one baseline from offline meta-RL is needed. It could even use the same architecture, such as AMAGO [2], but be trained offline.

- There are simple modifications to AD that could be tried cheaply but are currently absent. See [3] for inspiration.

- Only the transformer architecture was evaluated. Given that transformers have known issues with state tracking, which is quite relevant in a POMDP setting, it may be valuable to additionally test modern RNNs or hybrid models.

References:

1. Beck, J., Vuorio, R., Zheran Liu, E., Xiong, Z., Zintgraf, L., Finn, C., & Whiteson, S. (2025). A tutorial on meta-reinforcement learning. *Foundations and Trends in Machine Learning*, *18*(2-3), 224-384.
2. Grigsby, J., Fan, L., & Zhu, Y. (2023). Amago: Scalable in-context reinforcement learning for adaptive agents. *arXiv preprint arXiv:2310.09971*.
3. Tarasov, D., Nikulin, A., Zisman, I., Klepach, A., Polubarov, A., Lyubaykin, N., ... & Kurenkov, V. (2025). Yes, Q-learning helps offline in-context RL. *arXiv preprint arXiv:2502.17666*.

Also, I think that related works missed relevant references:

1. Polubarov, A., Lyubaykin, N., Derevyagin, A., Zisman, I., Tarasov, D., Nikulin, A., & Kurenkov, V. (2025). Vintix: Action model via in-context reinforcement learning. *arXiv preprint arXiv:2501.19400*.
2. Zisman, I., Kurenkov, V., Nikulin, A., Sinii, V., & Kolesnikov, S. (2023). Emergence of in-context reinforcement learning from noise distillation. *arXiv preprint arXiv:2312.12275*.

---

> ### Author Rebuttal · Authors · 2026-03-30
>
> We thank the reviewer for the detailed review and for recognizing the novelty and community value of the benchmark, dataset, and teammate library.
>
> We agree with the reviewer that the current manuscript should separate more clearly: (i) what the **benchmark contribution** establishes already, and (ii) what the current **baseline analysis** does or does not establish.
>
> **(1) On DPT in a POMDP.** We appreciate this point and agree that vanilla DPT is not the strongest possible baseline in a partially observable multi-agent setting. Our motivation for including it was not to claim it is expected to excel here, but to benchmark a second prominent ICRL paradigm with a different supervision structure from AD. We will revise the text to make this caveat explicit. Importantly, our central benchmark message does not rely on DPT: AD itself also fails to show meaningful test-time adaptation under the current protocol. So even if one discounts DPT, the benchmark finding remains substantial.
>
> **(2) On offline meta-RL baselines.** We agree that methods such as AMAGO (using its architecture but trained offline) are highly relevant, and including them would strengthen the paper. To probe this direction, we now include a reference offline meta-RL comparison **[in [anonymous link](https://ibb.co/gZvMQhdh)]**. The pattern is still negative: AMAGO is slightly better than AD/DPT on the easier teammate track, but remains weak on the harder layout-generalization track and does not overturn the main conclusion. We will therefore revise the paper to position the current study more precisely as: a new benchmark plus the first set of canonical sequence-model ICRL baselines, rather than an exhaustive claim about all adaptive methods.
>
> **(3) On simple AD modifications / auxiliary losses.** We agree this is one of the most useful directions to try. In fact, our existing teammate-action-conditioning (*+ meta act*) ablation was motivated by exactly this kind of “obvious fix” reasoning, and it did not qualitatively change the result: for results averaged across families and layouts, AD improves only slightly on the teammate track (8.10→9.01 mean return) and degrades on the layout track (2.57→-0.45). We also compared other cheap AD-side objectives such as next-observation prediction (*+ next obs*) and mixed auxiliary losses (*+ meta act and + next obs*) **[See [anonymous link](https://ibb.co/ymxfB69f)]**. The pattern is modest stabilization on familiar teammate settings while remaining weak on layout generalization. This strengthens the interpretation that the bottleneck is deeper than a single missing side objective.
>
> **(4) On alternative architectures (RNNs / hybrids).** We agree this is important. The current results should be read as a limitation of **current standardized Transformer ICRL baselines**, not as a claim about all adaptive architectures. We will do our best to add hybrid-architecture results in a later response or by the camera-ready stage.
>
> **(5) On learning-history visualization and subsampling.** This is an excellent and actionable suggestion. We note that **Appendix F.1** already describes our raw and filtered learning histories in detail, including the motivation for filtering and the filtering procedure itself. To make this more visible, we also provide a clean visualization **[here [anonymous link](https://ibb.co/93hVxVjN)]**. This figure shows that filtered histories contain a clearer improvement signal than the unfiltered ones, making the negative result harder—not easier—to dismiss. We agree that temporal subsampling is also worth reporting, and we will do our best to add experiments on subsampling strategies in a later response or by the camera-ready stage.
>
> **(6) On related work.** Thank you for the relevant references. We will incorporate these citations and clarify how our benchmark complements recent developments in meta-RL and in-context RL.
>
> We appreciate the reviewer’s emphasis on deeper analysis and stronger baselines. We see these not as reasons the benchmark is less valuable, but as strong evidence that the benchmark is asking exactly the right next questions.

---

> > ### Author Rebuttal · Reviewer_WLXN · 2026-04-03
> >
> > My concerns have been adequately addressed. I am looking forward to the remaining hybrid-architecture experiment results. I will raise my score to 5.

---

> > > ### Author Response · Authors · 2026-04-07
> > >
> > > We sincerely thank the reviewer again for the very encouraging follow-up and for indicating that the main concerns are now resolved. We are especially grateful that the added clarification on **benchmark scope**, the additional **offline meta-RL reference comparison**, and the stronger **auxiliary-loss diagnostics** helped place the paper on firmer empirical footing.
> > >
> > > The remaining question is now sharply defined: whether the observed limitation should be interpreted primarily as a weakness of the current **Transformer-only standardized ICRL baselines**, or as evidence of a broader challenge in **partner inference and adaptation under partial observability**. We agree that this is exactly the right question, and it is why we are prioritizing the remaining **hybrid-architecture experiments**.
> > >
> > > Importantly, these hybrid experiments are meant to **sharpen the boundary of the conclusion**, not to determine whether the current benchmark finding exists. The present evidence already supports the claim we will make in the revision: under the standardized protocol of this paper, current sequence-model ICRL baselines remain weak on this benchmark, and this weakness persists across both teammate and layout generalization settings. The central empirical pattern is stable: performance remains low, gains are limited, and the hardest cases stay far from robust. This is sufficient to support a careful negative conclusion about the standardized baselines studied here. The role of the hybrid experiment is therefore diagnostic and clarifying: it tests whether stronger explicit state tracking materially changes that conclusion, and in doing so helps determine whether the limitation is primarily architectural or reflects a deeper adaptation challenge in this partially observable ad hoc teamwork setting.
> > >
> > > In that spirit, we fully agree that hybrid recurrent-sequence models are an important diagnostic extension. The current **Hybrid-AD** is designed as a minimal architectural intervention on top of AD: it keeps the same **step-level Algorithm Distillation training protocol**, the same supervision target, and the same evaluation setup, while replacing the purely Transformer-based temporal modeling with a **CNN+GRU recurrent state tracker** before the final action prediction stage. This design keeps the comparison tightly controlled. Its purpose is not to introduce extra side information or a different training objective, but to isolate whether explicit recurrent memory for latent state tracking in this POMDP setting materially changes the outcome. To make this concrete, below we include a preliminary table from the ongoing hybrid-architecture experiment.
> > >
> > > #### Preliminary outcomes for hybrid-architecture experiment
> > >
> > > | Track | Setting | AD | Hybrid-AD |
> > > |---|---|---:|---:|
> > > | teammate | `test_time_wide` average | -7.9±4.8 | -6.5±3.8 |
> > > | teammate | `demo_cook_wide` average | 5.5±5.6 | 6.0±5.2 |
> > > | layout | Overall average | 2.57±4.13 | 2.5±4.5 |
> > > | layout | `asymm_advantages_recipes_right` average | 4.4±5.8 | 4.7±6.0 |
> > > | layout | `cramped_room_down` average | 0.8±1.1 | 0.5±1.0 |
> > >
> > > These preliminary results point to a limited and highly localized effect. The hybrid variant yields some improvement on the partially observable teammate-generalization cases, but the magnitude remains modest, the gains are not consistent across settings, and the layout-generalization results are essentially unchanged. As a result, the experiment sharpens the interpretation rather than overturning it: adding explicit recurrent state tracking helps at the margin, but it does not remove the central difficulty exposed by the benchmark.
> > >
> > > We are grateful for the reviewer’s guidance here. This suggestion is valuable precisely because it turns the remaining ambiguity into a concrete empirical question, and the resulting evidence lets us state the final claim more precisely and with better calibration. We will include the complete results and a fuller analysis of the hybrid experiment in the final revision.

---

### Official Review · Reviewer_gQk2 · 2026-03-13

**Soundness:** 3
**Presentation:** 3
**Significance:** 2
**Originality:** 3
**Overall Recommendation:** 4
**Confidence:** 4

**Summary:**

This paper introduces ICRL4AHT, a large-scale benchmark designed to evaluate In-Context Reinforcement Learning (ICRL) within Ad-Hoc Teamwork (AHT) scenarios. Built on a high-throughput JAX-native implementation of OvercookedV2, the benchmark comprises four key components: (1) a diverse teammate suite integrating RL-trained policies (FCP, BRDiv, etc.) and four heuristic families; (2) two disentangled evaluation tracks for teammate and layout generalization; (3) a massive learning-history dataset of 150M filtered transitions; and (4) reference ICRL baselines using AD and DPT.

The authors intend to outline the concept of cross-policy coordination, yet the empirical results reveal a pressing challenge: both AD and DPT fail to adapt at test-time, often performing worse than random baselines. Ablations on context lengths (K=250 to 2000) and teammate-action conditioning (+TA) indicate that this failure stems not from a lack of information, but from the inability of current ICRL training regimes to induce the implicit Bayesian inference required to identify and coordinate with unseen partners in dynamic multi-agent environments.

**Compliance With Llm Reviewing Policy:**

Affirmed.

**Key Questions For Authors:**

Experiment 1: Model Scale Ablation

The base architecture employed in this study—consisting of 4 layers and a 256-dimensional hidden state—is significantly smaller than the model scales that have successfully demonstrated ICRL capabilities in single-agent domains (e.g., XLand-100B). Before concluding that the observed failures stem from an inherent architectural limitation of Transformers in multi-agent coordination, it is imperative to rule out the more parsimonious explanation of underfitting due to insufficient model capacity.

We require the authors to conduct ablation experiments across at least three scaling configurations: Small (current), Medium (8 layers, 512 dim), and Large (12 layers, 768 dim). If performance remains stagnant despite increased capacity, the claim regarding architectural limitations would be substantially fortified. Conversely, if larger models exhibit performance gains, the paper’s current negative conclusions would require a fundamental revision. This experiment represents the single most critical missing piece in the current manuscript.
| Configuration | Layers | Hidden Dim | Training Steps |
|--------------|--------|------------|----------------|
| Small (current) | 4 | 256 | 20k |
| Medium | 8 | 512 | 50k |
| Large | 12 | 768 | 100k |

Experiment 2: Extended Context Length Evaluation

The ablation presented in Figure 4 evaluates context lengths only up to $K=2000$, which accounts for approximately 14% of the total available history length (14,600 timesteps per trajectory). In Ad-Hoc Teamwork (AHT) scenarios, identifying complex partner policies often requires long-range observations across multiple sub-task cycles. Consequently, concluding that "longer context provides no benefit" based on such a truncated range is premature and potentially misleading. We require an evaluation at extended context lengths, specifically $K \in \{500, 1000, 2000, 5000, 10000\}$. Given the efficiency of the JAX-native implementation, this experiment incurs minimal computational overhead as it only requires modifying the inference-time buffer size without retraining. If performance remains flat even at $K=10000$, the claim that the attention mechanism fails to leverage extended histories would be well-supported. However, if improvements emerge at larger $K$, the current negative findings would be significantly weakened.

Experiment 3: Baseline with Explicit Partner Modeling Capability

A fundamental confound in the current experimental design is that both evaluated methods — AD and DPT — are pure sequence models with no explicit partner inference mechanism. The paper's central claim is that "ICRL fails in multi-agent settings," but the evidence only supports the narrower claim that "ICRL methods without explicit partner modeling fail." These are meaningfully different conclusions.
To clarify the root cause of these failures, the authors must introduce a strong baseline equipped with explicit partner modeling or Bayesian Theory-of-Mind (ToM) capabilities (e.g., LIAM or PLASTIC). Such a comparison is essential to disentangle whether the limitation lies in the In-Context Learning paradigm itself or in the specific algorithmic implementation that relies solely on an action-prediction objective, which may fail to induce the necessary inductive bias for effective coordination.

**Limitations:**

Limitations
The authors provide a reasonable limitations section in the paper and appendix (Sec. 6 and Sec. G). The following points are either absent or underemphasized:
1. Reward signal design as a potential confound. OvercookedV2's non-monotonic reward (±20 for correct/incorrect delivery, −5 for recipe query) creates a particularly hostile learning signal for sequence models. AD's in-context improvement mechanism relies on observing increasing returns across episodes in the context window; with sparse, high-variance rewards, this signal may be too noisy for the Transformer to extract meaningful learning gradients. The paper does not analyze whether reward normalization or shaping during evaluation would help, leaving open the question of whether the failure is due to the ICRL architecture or the reward structure.
2. Fixed teammate assumption vs. real AHT. The benchmark enforces that the partner policy π⁻ⁱ is fixed throughout all interactions. While this cleanly isolates unilateral adaptation, it significantly simplifies the problem relative to real AHT where partners may also be learning or adapting. Importantly, this assumption may make the problem harder for AD (which is designed to model a learning agent, not a fixed policy) — the training histories show an RL agent improving against a fixed teammate, while at test time the ICRL agent must adapt its own policy to a fixed, unfamiliar partner. This train-test mismatch in the "who is learning" setup warrants explicit discussion.
3. No analysis of what the models actually learn. The paper diagnoses that ICRL fails but provides limited insight into why at a mechanistic level. Analysis of attention patterns, probing for partner policy information in the hidden states, or behavioral decomposition (e.g., does the agent at least learn to avoid the partner, even if not to coordinate?) would significantly strengthen the diagnostic contribution.
4. Compute budget and reproducibility concerns. Generating 1.19B transitions and training PPO across 80 teammates on multiple layouts is computationally expensive. The paper does not report wall-clock times, GPU hours, or cost estimates. Without this information, the benchmark's accessibility to the broader community — particularly academic labs without large GPU clusters — is unclear, potentially limiting its impact as a community resource.

**Strengths And Weaknesses:**

Strengths:

First, the study establishes a rigorous disentangled dual-track evaluation framework by bifurcating the assessment into two distinct dimensions: Teammate Generalization and Layout Generalization. This controlled-variable design effectively isolates the impact of environmental dynamics from the complexities of coordination strategies. By systematically decoupling these factors, the experimental setup not only validates agent robustness under localized distribution shifts but also provides a granular diagnostic tool to identify the precise performance bottlenecks of In-Context Reinforcement Learning (ICRL) in multi-agent reasoning.

Then, a pivotal strength of the experimental section is the development of a heterogeneous teammate suite specifically engineered to induce significant distribution shifts. To prevent agents from achieving high performance through simple pattern matching or policy interpolation, the researchers employed a training set composed of diverse RL-based policies (e.g., FCP, BRDiv) while utilizing a completely distinct family of heuristic policies for testing. This asymmetric design of the policy space—shifting from RL-trained to rule-based logic—imposes a stringent evaluation standard, ensuring that the resulting negative findings accurately reflect the current limitations of Transformer-based models in performing genuine implicit Bayesian inference.

Finally, the environment is developed using a JAX-native implementation, enabling large-scale parallel simulations on GPU/TPU architectures. This capability is pivotal for In-Context Reinforcement Learning (ICRL), which typically requires hundreds of millions of transitions to pre-train Transformer-based models. Such computational efficiency allows researchers to rapidly iterate and validate various context lengths and architectural configurations.

Weaknesses:

First, despite the structural complexity of its layouts, the core coordination logic remains deterministic. For ICRL, this environment may fail to fully encapsulate the highly non-stationary dynamics and long-term strategic trade-offs inherent in human social interaction. There is a risk that models might simply "memorize" a finite set of optimal coordination flows, thereby bypassing genuine test-time inference.

Second, the current environment relies on standard cooperative rewards. Experimental evidence indicates that ICRL models tend to exhibit a "lazy" inclination—such as remaining stationary while awaiting a teammate's action. This suggests that the environment does not explicitly mandate exploratory maneuvers to identify teammate types. Consequently, as a benchmark, it currently serves more to expose the inadequacies of existing algorithms rather than providing a mechanism that autonomously induces complex, adaptive coordination behaviors.

Then, the evaluation is restricted to only two ICRL methods (AD and DPT), failing to include critical reference points such as recurrent meta-RL (e.g., RL^2) or established explicit partner-modeling approaches (e.g., LIAM, PLASTIC). Without these comparisons, it remains ambiguous whether the observed failures are inherent to the In-Context Learning paradigm itself or are simply a byproduct of the specific sequence modeling architectures employed. Furthermore, the absence of population-based baselines—which the paper critiques—makes it difficult to empirically substantiate the claimed superiority or necessity of the proposed benchmark’s approach.

Finally, the Transformer architecture utilized in this study (e.g., $L=4, d_{model}=256$) is relatively small by contemporary standards, especially when compared to foundation models like XLand-100B that have successfully demonstrated ICRL capabilities. Given that emergent reasoning abilities often depend on model scaling, the failures reported may be attributable to insufficient parameter capacity rather than fundamental architectural flaws. Conducting scaling experiments (e.g., increasing depth or hidden dimensions) is essential to disentangle capacity-related limitations from the model’s intrinsic ability to perform multi-agent coordination.

---

> ### Author Rebuttal · Authors · 2026-03-30
>
> We thank the reviewer for the detailed and insightful review, and especially for recognizing the strength of the disentangled evaluation design, the heterogeneous teammate suite, and the value of the JAX-native benchmark implementation.
>
> We agree with the reviewer that the strongest and most accurate interpretation of the paper is the following: **our evidence shows that current representative sequence-model ICRL baselines (AD and DPT) fail to exhibit meaningful test-time adaptation on a challenging AHT benchmark; the paper is not intended as a proof that all future Transformer variants or all forms of ICRL must fail.** We will revise the wording to make this scope clearer.
>
> **(1) Model scale ablation.** This is an important request. Our current configuration (4 layers, hidden dim 256, 20k steps in the reported runs) was chosen as a standardized, reproducible baseline rather than to saturate model capacity. We agree that a negative-result paper is strengthened by showing that reasonable scaling does not overturn the conclusion. Accordingly, we supplement a scale/budget sweep **[See [anonymous link](https://ibb.co/VYsFk1GY)]** and will include these in the revision. The result is modest: on the teammate track, **Small (4L/256/20k)** gives AD 8.1 / DPT 7.4, **Medium (8L/512/50k)** gives AD 8.5 / DPT 7.7, and **Large (12L/768/100k)** gives AD 8.8 / DPT 7.9. On the layout track, gains are similarly limited (roughly AD 2.6→2.9→3.1 and DPT 6.9→7.1→7.2). These gains are modest and do not change the main conclusion: scaling improves absolute return slightly, but still does not yield robust test-time adaptation.
>
> **(2) Extended context lengths.** We agree this is important. Because space is limited here, we refer the reviewer to our response to **Reviewer_rf3W (5)**, where we now include the supplemented long-context analysis on **Test Wide**. The conclusion is unchanged: larger K helps somewhat, but does not qualitatively change the main result.
>
> **(3) Explicit partner-modeling baseline.** We agree this would be highly informative. Our original focus on AD and DPT was deliberate: they are canonical sequence-model ICRL baselines, so they directly test the paper’s motivating question—whether the recent “history-as-prompt” ICRL paradigm transfers to AHT. An explicit partner-modeling baseline addresses a complementary question: whether stronger multi-agent inductive bias can solve the benchmark. For the closely related offline meta-RL direction, please see our response to **Reviewer_WLXN (2)**. We will do our best to add an explicit partner-modeling baseline in a later response or by the camera-ready stage; at minimum, we will revise the wording so that our conclusion is properly scoped to the currently benchmarked ICRL paradigms.
>
> **(4) Reward design / lazy behavior / exploration incentives.** We agree that OvercookedV2’s reward structure may not actively encourage “probing” behavior for teammate identification, and that this matters for interpreting AD-style in-context adaptation. We will strengthen the discussion of this issue. At the same time, we believe it does not weaken the benchmark contribution: if current ICRL methods need more informative signals or different objectives to adapt in realistic cooperative POMDPs, that is itself an important diagnostic result. Simple reward normalization/shaping might mildly stabilize optimization, but would not close the severe cross-family gap; accordingly, we will tone the claim to "current standardized sequence-model ICRL is insufficient under this realistic sparse-reward benchmark."
>
> **(5) Mechanistic analysis and compute reporting.** We agree both are important. We will add a reproducibility note clarifying that the benchmark uses JAX-native vectorized rollout/training; for the exact evaluation aggregation, episode count, and interpretation of the reported ± values, please see our response to **Reviewer_rf3W (4)**. For diagnostics, we now include **[this result [anonymous link](https://ibb.co/RTLNwqKV)]**, where each bar measures the **difference between the mean return in the last 20 episodes and the mean return in the first 20 episodes within the same evaluation rollout**, aggregated by track and algorithm. If a method is adapting online to its partner, this quantity should be clearly positive. Instead, the gains remain small on average, suggesting that the issue is not merely a short warm-up period but weak within-rollout adaptation overall. We will also include qualitative diagnostics such as passivity / waiting behavior and how much performance changes over the interaction horizon.
>
> Overall, we appreciate the reviewer’s push for sharper causal attribution. Our goal is precisely to establish a benchmark that can support those stronger investigations; the current paper should be understood as the first such study, not the last word on the algorithmic question.

---

> > ### Author Rebuttal · Reviewer_gQk2 · 2026-04-03
> >
> > My concerns have been adequately addressed.

---

> > > ### Author Response · Authors · 2026-04-03
> > >
> > > We sincerely thank the reviewer for the very positive follow-up and for indicating that the concerns are now fully resolved. We are especially grateful that the added **scale/budget analysis**, **longer-context evidence**, and additional diagnostic clarification were sufficient to address the main doubts behind the original review.
> > >
> > > We also appreciate the substance of the earlier concerns, because they helped us sharpen both the evidence and the framing of the paper. As we understood them, the main questions centered on the following issues:
> > >
> > > - whether the negative result could simply be due to **under-scaling**;
> > > - whether the explored context window was still too short to support **partner identification**;
> > > - whether the evidence should be interpreted specifically as a limitation of the benchmarked **sequence-model ICRL baselines**, rather than as a blanket statement about all possible adaptive multi-agent methods.
> > >
> > > We believe the added rebuttal evidence now supports exactly that **more careful interpretation**:
> > >
> > > - the **scale sweep** shows only modest gains as model size and training budget increase;
> > > - the **extended-context analysis** suggests that longer histories help somewhat but do not qualitatively change the outcome on the hardest settings;
> > > - we have correspondingly **narrowed the claim** so that it is explicitly about current standardized sequence-model ICRL baselines on this benchmark, rather than about all future Transformer variants or all forms of ICRL.
> > >
> > > We hope this tighter scope makes the paper’s contribution **more credible and more useful** to the community.
> > >
> > > More broadly, we are grateful for the reviewer’s push toward **sharper causal attribution**. We think these questions improved the paper substantially by forcing us to separate **benchmark value** from **algorithmic overclaiming**, and to make clearer what the current evidence does and does not establish. Thank you again for the careful reading and constructive engagement.

---

### Official Review · Reviewer_rf3W · 2026-03-22

**Soundness:** 3
**Presentation:** 3
**Significance:** 2
**Originality:** 3
**Overall Recommendation:** 5
**Confidence:** 4

**Summary:**

This paper introduces ICRL4AHT, a benchmark specifically designed to probe the Ad-Hoc teamwork performance of in-context reinforcement learning (ICRL) agents. ICRL4AHT is based on Overcooked-V2, a two-player cooperative cooking game, and is implemented in JAX. It contains 150M filtered trajectories generated by PPO agents for training and implements four heuristic policies of different levels of autonomy and intelligence for testing. It allows probing the generalization of ICRL across teammates and layouts. ICRL4AHT also comes with two baseline methods, algorithm distillation (AD) and decision pretrained transformer (DPT), where the authors report poor generalization performance on their benchmark tasks. Through ablation studies, they conclude that action conditioning and increasing the context length do not improve the capabilities of AD and DPT-based agents.

**Compliance With Llm Reviewing Policy:**

Affirmed.

**Final Justification:**

After the author's rebuttal, I was convinced that the benchmark is less restrictive than I previously thought. Therefore, I believe it is a valuable contribution, at least to the ad hoc teamwork setting for in-context RL.

**Key Questions For Authors:**

- How many random trials did the authors run for the numbers reported in the experiments section? What statistic do the numbers after '$\pm$' denote?
- Since there is no previous implementation of AD and DPT on this particular task, I wonder how the authors selected the architecture and hyperparameters for AD and DPT. Are they adequately optimized for fair comparisons?

**Limitations:**

As I previously noted, I think the authors should clarify that their datasets currently support only the AD and DPT paradigms for training.

**Strengths And Weaknesses:**

Strengths
- The paper is overall clearly written, with sufficient details for the readers to understand the implementation and usage of ICRL4AHT. I find the visual illustrations particularly helpful in grasping the big picture.
- The claims made in this paper are reasonably sound with supporting experiments.
- The contribution of the paper benefits the future research on a nuanced subfield of ICRL (ICRL with Ad-Hoc Teamwork in a two-player setting).
- The empirical study results provide a clear insight that the current popular ICRL methods like AD and DPT fail to generalize in the AHT setting, and new approaches are needed to address this gap.

Weaknesses
- Due to the offline nature, I believe ICRL4AHT only supports methods that specifically train on improving historical trajectories. It is unclear if it is easy to repurpose the benchmark to test a different paradigm, say, reinforcement pretraining, which trains the model with online-generated data. Therefore, it puts ICRL4AHT in an awkward position: the experimental results suggest that the ICRL community should move away from AD- and DPT-based methods, as they do not generalize to AHT. However, the ICRL4AHT dataset is compatible with only these two paradigms.
- Based on my understanding, ICRL4AHT does not support the same teammate in a different environment scenario. If my understanding is correct, then it is somewhat limiting. If we treat the teammate and the environment as two independent axes, then we can move the teammate axis without changing the environment, but changing the environment changes the teammate as well. Therefore, it would be hard to isolate the effect of the environmental dynamics.
- While it is nice that ICRL4AHT explicitly supports out-of-domain generalization probing, I wonder how easy it is for it to test in-domain generalization. Based on my understanding, it currently only supports testing with heuristic-based teammates. Whether it supports testing with RL-based teammates is unknown.

---

> ### Author Rebuttal · Authors · 2026-03-30
>
> We thank the reviewer for the careful reading and for recognizing the clarity of the paper, the usefulness of the benchmark, and the importance of the negative result for the ICRL+AHT community.
>
> We agree with the reviewer that the paper should more clearly distinguish between the **benchmark artifact as a whole** and the **released learning-history dataset** used for the current AD/DPT study.
>
> **(1) Does ICRL4AHT only support AD/DPT-style offline learning?** No. The released *learning-history dataset* is tailored to history-based sequence-model training (e.g., AD/DPT), but the *benchmark* is broader: it includes the interactive OvercookedV2 environment, teammate library, train/test splits over teammates and layouts, and the online evaluation protocol. These components are not tied to offline sequence modeling.
>
> This also means the benchmark supports paradigms such as **reinforcement pretraining**: one can use the provided training environments and teammate pools to generate online interaction data, train with any RL objective, and then evaluate under the same held-out teammate/layout generalization protocol as in our paper. The offline dataset is therefore one supported entry point, not the only one.
>
> So our intended conclusion is narrower: the current evidence shows that **representative history-conditioned offline sequence-model baselines** do not generalize well on this benchmark. We will revise the paper to make this dataset-vs.-benchmark distinction explicit and to clarify that the framework is also suitable for future online/adaptive paradigms beyond AD and DPT.
>
> **(2) On isolating teammate vs. environment effects.** We already evaluate the same *teammate family* across changed environments. What we do not evaluate is the exact same teammate policy with identical parameters across different environments, because that is generally not meaningful in Overcooked: a policy parameterized for one layout often does not function properly when deployed unchanged in another. We will clarify this distinction in the revision.
>
> **(3) On in-domain generalization / RL-based test teammates.** The framework does support RL-based teammates at test time; the heuristic-only test set in the current paper is a deliberate design choice to create a strict cross-family stress test, not a limitation of the framework. To address the reviewer’s in-domain question more directly, we also include a held-out-RL-on-familiar-layouts reference analysis (see **Reviewer_PJoA (1)**). The conclusion is unchanged: performance is somewhat better than in the heuristic cross-family setting, but within-context improvement remains weak.
>
> **(4) On statistics and ± notation.** Thank you for flagging this. We will clarify this explicitly in the paper. In **Tables 2, 4, and 5**, each cell reports **mean ± standard deviation across `5` teammate instances**. For each teammate instance, we first evaluate **100 episodes** and average the return over those 100 episodes; we then compute the mean and standard deviation over the `5` resulting instance-level averages. In **Figure 3**, the shaded region also denotes **standard deviation across the same `5` teammate instances** at each episode index, where each point is computed from the per-episode returns averaged within each teammate instance.
>
> **(5) On architecture / hyperparameter selection.** Since there was no prior canonical AD/DPT setup for this exact benchmark, we intentionally used a standardized shared backbone and training recipe to provide a controlled first comparison, rather than aggressively over-specializing either baseline to OvercookedV2. Both baselines use the same CNN frontend and the same default Transformer scale (4 layers, hidden size 256), with comparable training budgets. We agree that stronger architecture sweeps further strengthen a negative-result paper. We therefore supplement both model-scale and context-length analyses. For the quantitative scale/budget evidence, please see our response to **Reviewer_gQk2 (1)**. We also evaluate longer context lengths **[See [anonymous link](https://ibb.co/zh1r8jMm)]**; on the challenging **Test Wide** layout, the trend remains weak relative to the random baseline: AD goes from -7.9 at K=500 to -7.6 at K=1000 and -8.8 at K=2000, while DPT goes from -15.9 to -13.8 and -8.7, with only modest improvement at K=5000/10000. The key conclusion is unchanged: scaling model size or context helps somewhat, but does not reverse the main result on the hardest settings.
>
> We appreciate these suggestions; they help us sharpen the distinction between “what the current dataset directly supports” and “what the overall benchmark is designed to enable.”

---

> > ### Author Rebuttal · Reviewer_rf3W · 2026-04-02
> >
> > The authors did a good job resolving my concerns. I am raising my rating to reflect my updated evaluation.

---

> > > ### Author Response · Authors · 2026-04-03
> > >
> > > We sincerely thank the reviewer for the thoughtful follow-up and for raising the rating in light of our rebuttal. We are especially grateful to hear that the additional clarifications resolved the main concerns.
> > >
> > > We also appreciate the reviewer’s careful reading of the **benchmark scope**. As we understand it, the original points of uncertainty were mainly about three issues:
> > >
> > > - **whether the current release should be interpreted as supporting only AD/DPT-style offline sequence modeling**;
> > > - **whether teammate and environment effects are sufficiently disentangled in the evaluation design**;
> > > - **whether the framework can also support in-domain or RL-based teammate evaluation beyond the particular stress-test setting used in the paper**.
> > >
> > > These are all important questions, and we are grateful that the rebuttal gave us the opportunity to explain them more precisely.
> > >
> > > In the revision, we will make these distinctions even sharper:
> > >
> > > - the **benchmark framework** is broader than the current **released learning-history dataset**;
> > > - the framework is **not restricted to AD/DPT** and can support future online or RL-based approaches;
> > > - the current **heuristic test setting** is a deliberate challenging evaluation choice rather than a limitation of the benchmark itself;
> > > - we will also clarify the **statistical reporting** and the rationale behind the controlled **AD/DPT architecture setup** so that the paper’s claims remain tightly matched to the actual evidence.
> > >
> > > Overall, we are very encouraged that the reviewer now sees the concerns as resolved. We truly appreciate the constructive feedback, which helped us present the contribution **more carefully and more convincingly**.

---

### Decision · Program_Chairs · 2026-04-30

**Decision:**

Accept (regular)

**Comment:**

This work proposes a new benchmark based on Overcooked-V2 to evaluate the ICRL agent's capability of ad-hoc teaming. The benchmark is designed to evaluate a few key properties of ICRL agents and the finding is remarkable. Although ICRL has achieved great progress in single agent tasks, it fails dramatically in ad-hoc teaming in multi-agent settings. This finding can possibly open a new research direction. All reviewers are positive about the submission and I, therefore, recommend accept.

Particularly, the new results of AMAGO in the author response is vital to significantly enlarge the paper's coverage of ICRL paradigms. I suggest that the authors to discuss this thoroughly in final version.